# AUTOAGENTS: A FRAMEWORK FOR AUTOMATIC AGENT GENERATION

## ABSTRACT

Large language models (LLMs) have enabled remarkable advances in automated task-solving with multi-agent systems. However, most existing LLM-based multi-agent approaches rely on predefined agents to handle simple tasks, limiting the adaptability of multi-agent collaboration to different scenarios. Therefore, we introduce AutoAgents, an innovative framework that adaptively generates and coordinates multiple specialized agents to build an AI team according to different tasks. Specifically, AutoAgents couples the relationship between tasks and roles by dynamically generating multiple required agents based on task content and planning solutions for the current task based on the generated expert agents. Multiple specialized agents collaborate with each other to efficiently accomplish tasks. Concurrently, an observer role is incorporated into the framework to reflect on the designated plans and agents' responses and improve upon them. Our experiments on various benchmarks demonstrate that AutoAgents generates more coherent and accurate solutions than the existing multi-agent methods. This underscores the significance of assigning different roles to different tasks and of team cooperation, offering new perspectives for tackling complex tasks. The repository of this project is available at https://anonymous.4open.science/r/AutoAgents-747C.

## 1 INTRODUCTION

Large language models (LLMs) have exhibited astounding capabilities as versatile task-solving agents, endowed with a rich blend of knowledge and skills. Nevertheless, they still face difficulties (Qin et al., 2023; OpenAI, 2023; Bubeck et al., 2023) in tackling various tasks that require intensive knowledge and reasoning, such as avoiding hallucination (Maynez et al., 2020), employing slow-thinking strategies (Sloman, 1996), ensuring trustworthiness (Wang et al., 2023a), and in combining diverse domain knowledge and long-horizon planning. In contrast, humans often exploit the benefits of collaborative problem solving, which enables them to work together effectively to solve non-routine problems in diverse domains and enhance the quality and reliability of the solutions by distributing the workload among specialties and applying a diversity of perspectives and expertise (Nelson, 2013; Roschelle & Teasley, 1995; Barron, 2000).

Inspired by collaborative problem solving, several recent works (Wang et al., 2023c; Du et al., 2023; Liang et al., 2023; Hao et al., 2023) have improved the task-solving capabilities of LLMs by integrating multi-agent discussion. However, most of these multi-agent systems depend on handcrafted or user-specified agents, with specific roles and necessitating human supervision, which often restricts the scope of collaborative applications. Moreover, manually creating a large number of experts often consumes a lot of resources. In order to adaptively solve more complex problems, this paper aims to explore a method of adaptively generating task experts and completing different tasks through multi-level collaborative cooperation among multiple experts.

In this paper, we propose AutoAgents, an innovative framework that adaptively generates and coordinates multiple specialized agents to construct an AI team according to different tasks. Figure 1 provides a high-level overview of AutoAgents. By generating multiple agents with distinct expert roles, we aim to form a collaborative entity that can accomplish complex tasks by leveraging the complementary strengths of each agent. As shown in Figure 2, the process of AutoAgents is divided into two critical stages: **Drafting Stage** and **Execution Stage**. The drafting stage involves a collaborative discussion among three predefined agents (**Planner**, **Agent Observer**, and **Plan**

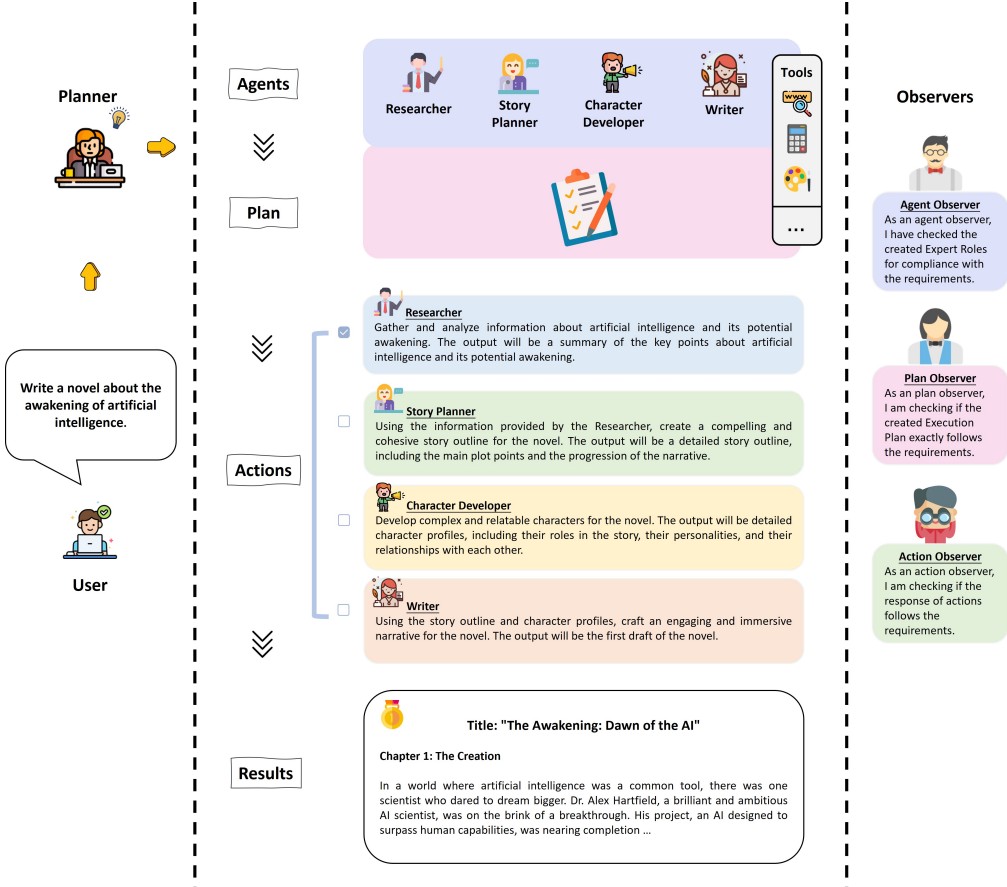

Figure 1: A schematic diagram of AutoAgents. The system takes the user input as a starting point and generates a set of specialized agents for novel writing, along with a corresponding execution plan. The agents collaboratively carry out the tasks according to the plan and produce the final novel. Meanwhile, an observer monitors the generation and execution of the Agents and the plan, ensuring the quality and coherence of the process.

**Observer**) to synthesize a customized agent team and an execution plan that suit the input problem or task. The execution stage refines the plan through inter-agent collaboration and feedback, and produces the final outcome. We propose self-refinement by individual agents and collaborative refinement by multiple agents to enhance agent proficiency and promote knowledge-sharing among agents. To facilitate the specific division of labor among the agents in the synthesized team, we introduce a predefined agent (**Action Observer**) to assist the agent team in sharing information, coordinating actions, reaching consensus, and adapting to the environment.

To synthesize heterogeneous information from diverse domains is often a crucial requirement in creative industries and other real-world scenarios. We illustrate a concrete example of how AutoAgents tackles the challenging task of writing a novel about the awakening of artificial intelligence in Figure 1. The Story Planner and Researcher collaborate to devise the plot of the story with their respective expertise, while the Character Developer and Writer enrich the novel content through imagination based on the story. Moreover, we conduct quantitative experiments and case studies in complex tasks to demonstrate the effectiveness of AutoAgents. We also conduct a comprehensive analysis and demonstrate the importance of dynamic agents for handling complex tasks, the indispensability of self-refinement for proficient agents, and the effectiveness of collaborative conversation.

To summarize, this paper makes the following novel contributions: **First**, we propose AutoAgents, a novel framework that dynamically synthesizes and coordinates multiple expert agents to form customized AI teams for diverse tasks. **Second**, we conduct rigorous quantitative experiments on two challenging tasks and demonstrate that AutoAgents significantly improves both knowledge acquisition and reasoning ability in LLMs and outperforms other generated-agent frameworks. **Third**,

Table 1: Comparison of existing and proposed frameworks for LLM-based Agent framework.

| Framework | Dynamic Agent Generation Method | Number of Agent | Multi-agent Conversation | Self-Refinement Action | Collaborative Refinement Action |
|---|---|---|---|---|---|
| AutoGPT (Gravitas, 2023) | ✗ | 1 | ✗ | ✓ | ✗ |
| BabyAGI (Nakajima, 2023) | ✗ | 3 | ✓ | ✗ | ✗ |
| Generative Agents (Park et al., 2023) | ✗ | 25 | ✓ | ✓ | ✗ |
| Camel (Li et al., 2023) | ✗ | 2 | ✓ | ✗ | ✗ |
| GPT-bargaining (Fu et al., 2023) | ✗ | 3 | ✓ | ✓ | ✗ |
| MetaGPT (Hong et al., 2023) | ✗ | Unlimited | ✓ | ✗ | ✗ |
| AutoGen (Wu et al., 2023) | ✗ | Unlimited | ✓ | ✗ | ✗ |
| Social Simulacra (Park et al., 2022) | Single Agent | Unlimited | ✓ | ✗ | ✗ |
| Epidemic Modeling (Williams et al., 2023) | Single Agent | Unlimited | ✓ | ✗ | ✗ |
| ExpertPrompting (Xu et al., 2023) | Single Agent | 1 | ✗ | ✗ | ✗ |
| SSP (Wang et al., 2023c) | Single Agent | Unlimited | ✓ | ✗ | ✗ |
| AgentVerse (Chen et al., 2023a) | Single Agent | Unlimited | ✓ | ✗ | ✗ |
| **AutoAgents** | Multi-agent Discussion | Unlimited | ✓ | ✓ | ✓ |

we showcase AutoAgents' ability to adapt to complex tasks by applying it in various scenarios such as software development. **Finally**, we conduct a thorough investigation and reveal the importance of dynamic agents for accommodating complex tasks and the necessity of self-refinement for proficient agents, and the efficacy of collaborative conversation.

## 2 RELATED WORK

**LLM-based Autonomous Agents.** LLMs have been widely used as core controllers for autonomous agents that can accomplish specific objectives. Auto-GPT (Gravitas, 2023) is an early work that leverages an LLM as an AI agent that can autonomously achieve a given goal with the help of several tools. However, Auto-GPT does not support multi-agent collaboration and can only work in isolation. One way to enhance the task-solving capabilities of LLMs is to assign different roles and responsibilities to multiple LLMs and let them coordinate their actions to achieve a common goal. For example, BabyAGI (Nakajima, 2023) is an AI-powered task management system with multiple LLM-based agents. One agent creates new tasks based on the previous task's objective and result, another agent prioritizes the task list, and another agent completes tasks. BabyAGI is a multi-agent system with a fixed order of agent communication. MetaGPT (Hong et al., 2023) is a multi-agent framework for assigning different roles to GPTs to form a collaborative software entity for complex tasks. It is a specialized LLM-based multi-agent framework for collaborative software development. Camel (Li et al., 2023) is an LLM-based communicative agent framework that demonstrates how role-playing can be used to enable chat agents to communicate with each other for task completion. However, Camel does not support tool-using. Several recent works (Wang et al., 2023c; Du et al., 2023; Liang et al., 2023; Hao et al., 2023) have enhanced the task-solving capabilities of LLMs by integrating multi-agent discussion. For instance, Wang et al. (2023c) proposes a multi-agent debate system that allows LLMs to argue for or against a given claim and generate a debate summary. Du et al. (2023) introduce a multi-agent dialogue system that enables LLMs to exchange information and opinions on a given topic and generate a dialogue report. AutoGen (Wu et al., 2023) is a framework that enables the development of LLM applications using multiple agents that can converse with each other to solve tasks. However, most of these multi-agent systems rely on handcrafted or user-specified agents with specific roles and do not support the automatic generation of agents, which often limits the scope of collaborative applications.

**Agent Generalization.** Several studies (Park et al., 2022; Williams et al., 2023) employ LLMs to generate agents for social simulacra and epidemic modeling, demonstrating how this technique can facilitate designers in assessing and improving their modeling designs prior to deploying them to real users. Likewise, ExpertPrompting (Xu et al., 2023) devised a method to generate diverse profiles of agents that can cooperate with human users to accomplish tasks with minimal supervision. However, this method still depends on a restricted set of predefined agents, and the generated agents vary only in their profiles, not in their problem-solving capacities. Recently, SSP (Wang et al., 2023c) and AgentVerse (Chen et al., 2023a) have proposed frameworks for automatically generating unlimited agents. SSP enables LLMs to generate agents for problem input by providing some agent samples, and has these agents solve the problem. AgentVerse generates the execution plan through the generated agents' discussions and adds evaluation strategies for cyclic execution. Unlike the previous two methods, AutoAgents places a heightened emphasis on the reliability of its generated agents and

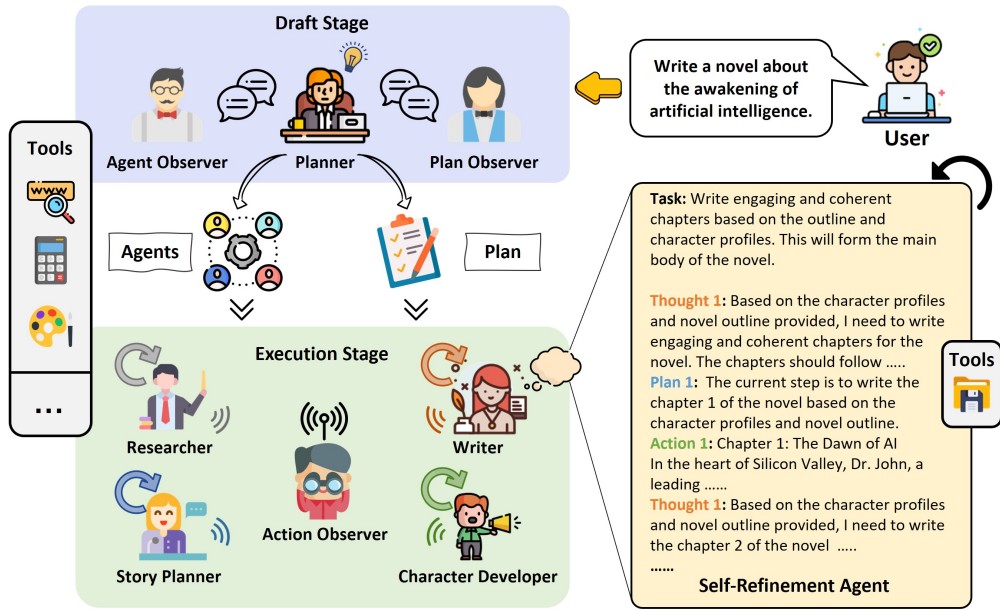

Figure 2: The execution process of AutoAgents. During the **Drafting Stage**, three predefined agents collaboratively determine the list of agents and the execution plan. During the **Execution Stage**, a predefined agent facilitates coordination and communication among the generated agent teams, and the individual generated agents enhance their execution efficiency through self-refinement.

strategic plans, thereby enhancing task execution efficiency through the utilization of collaborative refinement actions and the integration of self-refinement actions , as illustrated in Table 1.

# 3    THE FRAMEWORK FOR AUTOMATIC AGENT GENERATION

To enhance the effectiveness of autonomous multi-agent groups in accomplishing their goals, the process of AutoAgents consists of two critical stages: **Drafting Stage** and **Execution Stage**, as illustrated in Figure 2. The drafting stage synthesizes an agent team and an execution plan that are customized to the task by analyzing the input problem or task. The execution stage refines the plan by enabling inter-agent collaboration and feedback, and delivers the final result. The inter-agent collaboration is based on some principles of multi-agent cooperation, such as communication, coordination, and consensus. These principles help the agents to share information, align their actions, reach agreements, and adapt to the environment.

## 3.1    DRAFTING STAGE

Empirical evidence  (Woolley et al., 2015) suggests that diversity within human groups fosters diverse perspectives, which enhances the group's performance across various tasks. The drafting stage, which determines the composition of a multi-agent group, plays a crucial role in setting the upper limits of the group's capabilities. Therefore, it is imperative to generate the optimal agent team and execution plan that can maximize the group's potential.

Predominant methodologies (Gravitas, 2023; Hong et al., 2023; Wu et al., 2023) for assigning role descriptions to autonomous agents rely heavily on human intuition and prior knowledge, requiring manual assignment based on task understanding. Consistent with several parallel findings (Xu et al., 2023; Wang et al., 2023c; Chen et al., 2023a), dynamically designing agents with different roles can significantly enhance their efficacy. However, the scalability and rationality of agent and plan generation are still unclear, especially in the face of various complex problem environments.

On the one hand, the generated agents should exhibit diversity to accommodate various tasks. On the other hand, the agent and the plan generation should adhere to certain principles, rendering their role allocation more rational. Therefore, we devise three artificially predefined agents to produce agent teams and execution plans, integrating artificial prior knowledge and the dynamic adaptation

capability of LLMs to generate more sensible agent teams and execution plans. The three artificially predefined agents comprise **Planner**, **Agent Observer**, and **Plan Observer**:

- **Planner** $\mathcal{P}$ generates and refines an agent team and an execution plan based on the content of the task.

- **Agent Observer** $\mathcal{O}_{agent}$ provides suggestions on the rationality of the agent team members and their matching degree with the task.

- **Plan Observer** $\mathcal{O}_{plan}$ provides suggestions on the rationality of the execution plan and its matching degree with the task and the agent team.

The Planner generates initial agent team members and a specific plan, and improves the agent team and execution plan based on continuous communication with the Agent Observer and Plan Observer.

**Agent Generation.** The Planner generates the agent team and facilitates its continuous improvement through reciprocal communication with the Agent Observer. To enable Planner to produce rational agents, we have devised a standard format for the essential elements of a single agent. For each agent $\mathcal{A} = \{P, D, T, S\}$, the Planner needs to specify its prompt P, description D, toolset T, and suggestions S.

- **Prompt** P provides a detailed and customized depiction of the expert identity for each specific agent, which comprises profile, goal, and constraints. **Profile** reflects the domain expertise of the role or job title. **Goal** indicates the primary responsibility or objective that the role aims to achieve. **Constraints** specify limitations or principles the role must adhere to when performing actions.

- **Description** D gives additional concrete identity to help establish a more comprehensive role, develop an execution plan, and inspect problems.

- **Toolset** T equips the Agent with tools that it can use, selected from a predefined set of tools. The rationale for not using all the tools for each agent here is to prevent decision-making confusion caused by excessive tools.

- **Suggestions** S supplies some suggestions for each agent to execute the current task, including but not limited to a clear output, extraction of historical information, and suggestions for execution steps.

Based on the agent list $\{\mathcal{A}_1, \mathcal{A}_2, \cdots, \mathcal{A}_n\}$ generated by Planner, the Agent Observer evaluates the quality and suitability of each agent. The Agent Observer first verifies whether every agent conforms to the aforementioned specifications and identifies any missing elements $\{P, \text{description } D, \text{toolset } T\}$. Secondly, the Agent Observer assesses the compatibility of each agent with the task, according to their description information and task content. Finally, the Agent Observer examines the agent list for any redundant or missing roles and eliminates or adds them accordingly.

After $n$ rounds of bidirectional communication between the Planner and the Agent Observer, the optimal agent list for accomplishing the task is established. Given the vital role of the agent list in the task execution, this framework employs a predefined agent and multiple rounds of iterative dialogue among multiple agents to finalize the agent list, thereby enhancing the stability and reliability of the execution phase.

**Plan Generation.** In parallel to agent generation, the Planner formulates the execution plan and promotes its progressive improvement through reciprocal communication with the Plan Observer. For a given task, the Planner delineates the specific steps $\{\mathcal{S}_1, \mathcal{S}_2, \cdots \mathcal{S}_n\}$ to accomplish it in the execution plan $\mathcal{P}$. Each step $\mathcal{S}_i$ entails a clear identification of the agent $\mathcal{A}_j$ responsible for it, as well as the input information and expected output required for it.

The Plan Observer subsequently validates the execution plan $\mathcal{P} = \{\mathcal{S}_1, \mathcal{S}_2, \cdots \mathcal{S}_n\}$ according to the agent list $\{\mathcal{A}_1, \mathcal{A}_2, \cdots, \mathcal{A}_n\}$ and the task content. It firstly ensures that each step has a corresponding agent and that the step content is coherent and concise. It secondly assesses whether all the steps are sufficient, whether the task can be accomplished, and whether there are any gaps that need to be filled. It finally provides feedback to the Planner, who further refines the execution plan accordingly. After $n$ rounds of dialogue between the Planner and the Plan Observer, the ultimate execution plan for achieving the task is established.

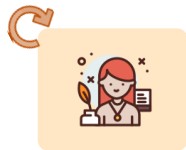

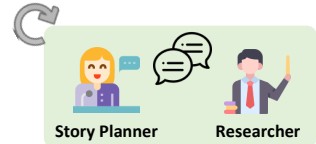

**Task:** Write engaging and coherent chapters based on the outline and character profiles. This will form the main body of the novel.

**Task:** The Story Planner collaborates with the Researcher to understand AI concepts and create a detailed outline for the novel. This includes a high-level overview of the story, a breakdown of the story into chapters, and a breakdown of each chapter into scenes.

(a) Self-refinement by a single agent      (b) Collaborative refinement by multiple agents

Figure 3: Two types of actions for executing tasks: **Self-refinement** enables an individual agent to enhance its competence in performing some specialized tasks. **Collaborative refinement** facilitates knowledge exchange among multiple agents and accomplishes tasks that demand interdisciplinary expertise.

**Task Execution Actions.** The Planner devises an execution plan that automatically assigns the requisite agents for diverse tasks. The execution plan comprises two actions of task execution: self-refinement by a single agent and collaborative refinement by multiple agents, as shown in Figure 3. Self-refinement empowers an individual agent to augment its proficiency in accomplishing some specialized tasks. Collaborative refinement fosters knowledge sharing among multiple agents and achieves tasks requiring interdisciplinary expertise.

## 3.2 EXECUTION STAGE

In the drafting phase, the framework generates an agent list and an execution plan based on the task requirements. Then, the framework creates corresponding roles and executes the plans in the execution environment[1]. The communication and cooperation among multi-agent systems are essential for accomplishing the tasks effectively. This section elaborates on the communication among multiple agents, the task execution strategies, and the knowledge-sharing mechanisms.

**Communication of Multiple Agent.** The communication structures among agents have been investigated by many studies (Chen et al., 2023a; Wang et al., 2023c; Qian et al., 2023; Chan et al., 2023) to examine their impact on task performance. In this framework, we adopt the vertical communication paradigm, which assigns different responsibilities to agents according to their roles. To facilitate the specific division of labor among the agents in the generated team, we introduce a predefined **Action Observer** as the team leader to coordinate the execution plan. Specifically,

- **Action Observer** $\mathcal{O}_{action}$ acts as the task manager for the different agents, allocating different tasks to them, verifying the execution outcomes of each agent, and dynamically adapting the execution plan based on the execution status.

This mechanism of refinement and communication recurs until the Action Observer attains a unanimous agreement on the execution responses, or the process reaches its maximum iteration limit. For scenarios that demand iterative decision-making towards specific objectives, such as software development, vertical communication would be a preferable option.

**Self-refinement Agent.** Besides the inter-agent communication, the performance of a single agent also exerts a significant impact on the overall quality of feedback results. Hence, drawing on mechanisms such as AutoGPT (Gravitas, 2023) and ReAct (Yao et al., 2022), we have devised a self-refinement mechanism for an individual agent.

For a single agent $\mathcal{A}$, the action at step $t$ is at $a_t = l_t \cup p_t \cup o_t$, where $l_t$ denotes the *thought* or the *reasoning trace* in the language space, which does not alter the external environment, and thus yields no observational feedback, $p_t$ represents the execution plan for better task completion, $o_t$ comprises the completion steps and execution output for this time.

As illustrated in Figure 2, various types of useful thoughts can assist in devising a refinement plan. The execution plan enables the agent to anticipate the steps they need to undertake in the future,

---

[1]Execution environment of AutoAgents is built based on MetaGPT's environment and workspace (Hong et al., 2023).

and the observational content of the execution result construction allows the agent to reevaluate and enhance the plan arrangement, thereby constructing more refined and complete actions. Through a cycle of self-continuous thinking, planning, execution, and feedback, a single agent can effectively execute and accomplish task content.

**Collaborative Refinement Action.** In the collaborative refinement action, the agents collaboratively refine and execute the tasks in a sequential manner. Each round of the collaboration involves a fixed order of turn-taking among the agents, who generate their responses based on the current observation. The chat history slot of each agent is updated by concatenating the previous utterances of the other agents. The collaboration terminates automatically when the agents reach a consensus or the maximum number of discussions is reached.

**Knowledge Sharing Mechanism.** AutoAgents also facilitates the sharing of execution results among various agents for improved communication and feedback. However, when the number of agents is large and a single agent has more self-iterations, it will generate more historical information. Due to the token limitation of LLM models, they often cannot encompass all information. Hence, this framework provides short-term memory, long-term memory, and dynamic memory.

*Short-term memory* is chiefly concentrated on a singular action, encompassing the gamut of intermediary notions, strategies, and outcomes that emerge during the self-refinement or collaborative refinement phases of an individual action. It is salient to note that these actions frequently culminate in a distilled summary of critical information, epitomizing the final phase of the refinement trajectory.

*Long-term memory* principally focuses on chronicling the historical trajectory of multifarious actions, predominantly documenting the executed tasks of individual actions along with the synthesis of vital feedback information. This aspect is imperative for evaluating the comprehensive extent of task completion.

*Dynamic memory* predominantly serves actions necessitating specialized attention. The Action Observer, having access to long-term memory archives, adeptly extracts ancillary information, dynamically tailoring it to the specific requirements of the action for task execution. This process significantly augments the efficiency of a single action in task fulfillment.

## 4 EXPERIMENTS

In order to demonstrate the capabilities and performance of AutoAgents in orchestrating autonomous agent groups to collaboratively accomplish tasks, we have performed extensive quantitative experiments on benchmark tasks and thorough case studies on more complex and realistic applications. In the quantitative analysis, we mainly present results for the **Open-ended Question Answer** task (detailed in Section 4.1) and the **Trivia Creative Writing** task (detailed in Section 4.2) to evaluate the framework effectiveness under distinct settings. The **Case Studies**, discussed in Section 4.3, illustrate the potential of a multi-agent group tackling intricate practical scenarios cooperatively.

**Implementation Details:** We conduct all experiments using the GPT-4 API[2] and set the temperature to 0 to ensure reproducibility. The rationale behind this selection is the exceptional performance these models offer, providing more accurate and standardized output. Additionally, their accessibility and ease of use through APIs enable us to directly call and interact with the models during our research, significantly simplifying the process. The maximum number of discussions during the drafting phase is 3, and the maximum number of self-refinement by a single agent and collaborative refinement by multiple agents during the execution phase is 5.

### 4.1 OPEN-ENDED QUESTION ANSWER

**Task Description.** Open-ended Question Answering is a crucial and challenging task in the domain of NLP and generative AI. It requires an AI system to produce coherent, elaborate, and human-like responses to questions that have no predetermined or fixed set of possible answers. (Zheng et al., 2023) proposed MT-bench, a benchmark consisting of 80 high-quality collected open-ended questions from various categories such as common sense, counterfactual, coding, etc. We then utilize

---

[2]The specific model version employed is "GPT-4-0613".

AutoAgents to produce collaborative answers based on multiple generated agents and compare them with the responses given by **GPT-4** and **AgentVerse** [3] (Chen et al., 2023a).

Table 2: Win Rate of AutoAgents over other models on Open-ended Question Answer, with FairEval (Wang et al., 2023b) and HumanEval serving as evaluators.

| Evaluator | v.s. **GPT-4** | v.s. **AgentVerse** |
|---|---|---|
| FairEval (Wang et al., 2023b) | 76.3% | 91.3% |
| HumanEval | 62.5% | 77.5% |

**Evaluation Metrics.** To measure the quality of open-ended responses with minimal evaluation bias, we adopt **FairEval** (Wang et al., 2023b) and HumanEval as the evaluation metrics for both the single agent and AutoAgents. FairEval incorporates several methods to mitigate the impact of various sources of bias, resulting in a better alignment with human judgment. For **HumanEval**, we enlisted three independent volunteers to evaluate two sets of responses—one generated by AutoAgents and the other by a different model—based on criteria such as helpfulness, reliability, accuracy, and comprehensiveness. Notably, the volunteers were blinded to the identity of the model that produced each response, ensuring an unbiased assessment. Appendix B contains the detailed scoring criteria for the ratings.

**Results.** Table 2 demonstrates that AutoAgents outperforms individual LLM models in both FairEval based on LLM and Human evaluations. AutoAgents can produce more comprehensive and nuanced answers to open questions by synthesizing multiple expert models. It can also provide more elaborate explanations and justifications for its answers. Additionally, AutoAgents demonstrates superior performance over AgentVerse. This enhanced efficacy is attributed in part to the reliability of agent generation, self-refinement, and collaborative refinement capabilities within AutoAgents. Conversely, AgentVerse necessitates additional task-specific adaptations and exhibits limited effectiveness in adapting to open-ended questions. More examples are given in the Appendix A.

## 4.2 TRIVIA CREATIVE WRITING

**Task Description.** The Trivia Creative Writing task (Wang et al., 2023c) challenges the capabilities of large language models to retrieve and integrate diverse information from their internal self-compressed knowledge. This task requires a model to craft a coherent story around a given topic while incorporating the answers to $N$ trivia questions. We evaluate the models under two settings, $N = 5$ and $N = 10$, where a higher $N$ entails more trivia questions and thus demands the model to exhibit more extensive domain knowledge. We constructed a benchmark consisting of 100 instances for each $N$, encompassing a total of 1000 trivia questions.

Table 3: The results of Trivia Creative Writing task. $\Delta$ indicates the differences compared with Standard Prompting (first row).

| Methods | N (# trivia questions) = 5 | | N (# trivia questions ) = 10 | |
|---|---|---|---|---|
| | Score (%) | $\Delta$ (v.s Standard %) | Score (%) | $\Delta$ (v.s Standard %) |
| Standard | 74.6 | 0.0% | 77.0 | 0.0% |
| CoT (Yao et al., 2023) | 67.1 | -10.0% | 68.5 | -11.1% |
| SPP-Profile (Wang et al., 2023c) | 79.1 | +5.9% | 83.0 | +7.8% |
| SPP (Wang et al., 2023c) | 79.9 | +7.1% | 84.7 | +10.0% |
| **AutoAgents** | **82.0** | **+9.9%** | **85.9** | **+11.6%** |

**Evaluation Metrics.** Drawing on the approach of (Wang et al., 2023c), we adopt an automatic metric to identify factual errors and measure a model's capacity to integrate diverse domain knowledge. We conduct string matching with the veridical target answers for each question on the generated output. The target answers are supplied from the TriviaQA dataset (Joshi et al., 2017), and each question can have a list of answer variants. A match to any of the answer variants of a question is regarded as a correct mention. The metric score is calculated as Trivia Creative Writing Metric Score = # correct answer mentions/# trivia questions.

---

[3]The prompt configuration within AgentVerse is tailored to its *brainstorming* task, incorporating modifications to upgrade the model to GPT-4.

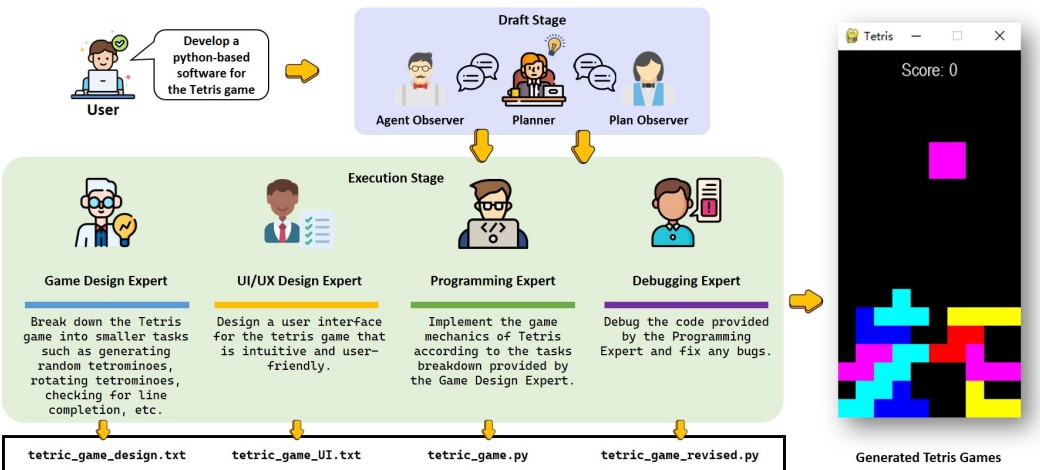

Figure 4: The illustration of an example process of software development. The task is to *develop Python-based software for the Tetris game*.

**Results.** Table 3 demonstrates the superior performance of AutoAgents in knowledge acquisition over the existing methods. Compared to the Standard method, which does not employ Agent Generation, AutoAgents achieves a remarkable 10% improvement across all experiments. Moreover, AutoAgents also surpasses SSP (Wang et al., 2023c), which utilizes agent generation but with a different approach. The enhanced performance of AutoAgents can be attributed to its elaborate methods of agent generation discussions and task execution including collaborative refinement and self-refinement. More examples are given in the Appendix A.

## 4.3 Case Study

To demonstrate the applicability of AutoAgents to more sophisticated and realistic scenarios, we conduct a case study in the software engineering domain. Software engineering is a complex collaborative endeavor that involves diverse roles and responsibilities. From developers who create the underlying code, to UI designers who prioritize user experience, and software testers who ensure the software's quality, experts collaboratively work to enhance and refine the application, ensuring that it meets both functional and user-centric criteria.

As an illustration in Figure 4, a Tetris game has been developed by employing AutoAgents, which has generated various expert roles, such as game design expert, UI design expert, programmer, and debugging expert, to accomplish the game development task. The game design experts provide the game logic documents that specify the rules and mechanics of the game. The UI design experts design the UI components that create the visual interface of the game. The programmers implement the game design based on the aforementioned documents and use appropriate programming languages and tools. Finally, the debugging expert tests the game and debugs the program to ensure its functionality and quality. The game development process is based on the collaboration of multiple expert roles, with more elaborate documentation and programs, making it easier for users to comprehend.

## 5 Conclusion

This paper introduces AutoAgents, an innovative framework for automatically synthesizing collaborative specialized agents. AutoAgents mimics the collaborative process of human teams by decomposing tasks into drafting and execution phases and delegating different subtasks to different agents. Our experimental and empirical evaluation validates the advantages of AutoAgents, as it surpasses single agents and other groupings in various tasks that demand diverse skills. Furthermore, our case study in software development illustrates the versatility and potential benefits of our proposed framework. AutoAgents opens up new possibilities for enhancing the interaction and cooperation among agents and transforms the landscape of complex problem-solving. We envisage that its principles can be further generalized and refined to deal with a broader range of tasks, paving the way towards more useful assistive artificial intelligence.

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

## A  FURTHER ANALYSIS

In order to further analysis the importance of various components in AutoAgents, this section will conduct a separate analysis of the *self-refinement action*, *collaborative refinement action*, *dynamic memory*, and *observers in the draft stage* within AutoAgents on 20 instances [4] of Trivia Creative Writing task and more case study.

Table 4: The ablation studies of AutoAgents on 20 instances of Trivia Creative Writing task. Δ indicates the differences compared with Standard Prompting (first row).

| Methods | N (# trivia questions) = 5 | |
|---|---|---|
| | Score (%) | Δ (v.s Standard %) |
| Standard | 74.6 | 0.0% |
| CoT (Yao et al., 2023) | 66.0 | -11.5% |
| SPP-Profile (Wang et al., 2023c) | 74.0 | -0.01% |
| SPP (Wang et al., 2023c) | 84.4 | +13.1% |
| **AutoAgents** w/o observers | 87.0 | **+16.6%** |
| **AutoAgents** w/o self-refinement | 87.0 | **+16.6%** |
| **AutoAgents** w/o collaborative refinement | 88.0 | **+18.0%** |
| **AutoAgents** w/o dynamic memory | 89.0 | **+19.3%** |
| **AutoAgents** | **90.0** | **+20.6%** |

**Collaborative discussion is crucial for rational agent generation and plan allocation.** During the Drafting Stage, the Planner in AutoAgents engages in collaborative discussions with two Observers to determine the optimal list of agents and the execution plan. Figure 5 illustrates the contrast between agent generation with and without collaborative discussion. In the absence of Observer feedback, the Planner tends to generate programmers exclusively to accomplish game development, neglecting the holistic process of game creation. With the input and coordination of the Observers, the Planner incorporates game design experts, UI design experts, and testing experts into the agent list. It is evident that the agent generation under collaborative discussions is more comprehensive and more aligned with the realistic scenarios of game development. This also corroborates the significance of collaborative discussions for agent generation and plan allocation, which will subsequently influence the execution outcomes.

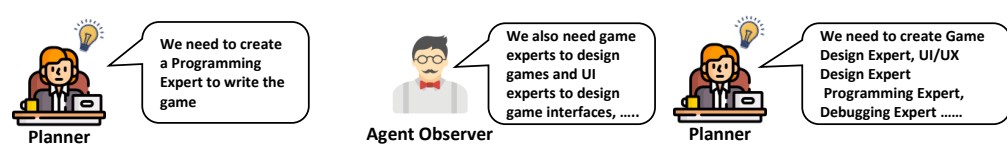

(a) w/o Collaborative Discussion    (b) w/ Collaborative Discussion

Figure 5: Comparison of whether there is a collaborative discussion in the Drafting Stage in the task that *developing Python-based software for the Tetris game*.

---

[4]Choose the final 20 samples from a dataset comprising 100 samples to serve as test samples.

Concurrently, Table 4 elucidates that in the absence of observers, there is a marked 3% reduction in the overall performance of AutoAgents. This substantiates the imperative role of collaborative discussions in agent generation. As depicted in Table 5, AutoAgent markedly enhances the caliber of agent generation via collaborative discussions, a facet notably overlooked by other generative frameworks in their consideration of agent generation quality. The empirical data presented in Table 2 and 3 further accentuate the superiority of AutoAgents when juxtaposed against counterparts like AgentVerse and SPP.

Table 5: Comparison of existing and proposed frameworks for multi-agent generation methods.

| Framework | Application | Agent Generalization by Multi-Agent Discussion | Prompt Generalization |
|---|---|:---:|:---:|
| Social Simulacra (Park et al., 2022) | Social Simulation | ✗ | ✗ |
| Epidemic Modeling (Williams et al., 2023) | Social Simulation | ✗ | ✗ |
| SSP (Wang et al., 2023c) | General Autonomous Agents | ✗ | ✗ |
| AgentVerse (Chen et al., 2023a) | General Autonomous Agents | ✗ | ✗ |
| **AutoAgents** | General Autonomous Agents | ✓ | ✓ |

**Enhancing single-agent through self-refinement.** Self-Refinement (Madaan et al., 2023; Shinn et al., 2023; Gou et al., 2023; Chen et al., 2023b; Huang et al., 2022; Yao et al., 2022) is a technique that enables LLMs to "converse" with themselves, evaluate their own generation, and iteratively improve their answers. Self-refinement has been shown to enhance the accuracy of LLMs' outputs in various domains (Madaan et al., 2023; Shinn et al., 2023; Gou et al., 2023; Chen et al., 2023b; Huang et al., 2022; Yao et al., 2022). Although AutoAgents is a framework for multi-agent collaboration, it also requires self-refinement agents to perform specialized roles for individual tasks.

Figure 6 depicts the self-refinement process of programmers' coding. They first write a pseudo code file and then generate the corresponding program files based on it. This refinement process significantly ensures the validity of the output file.

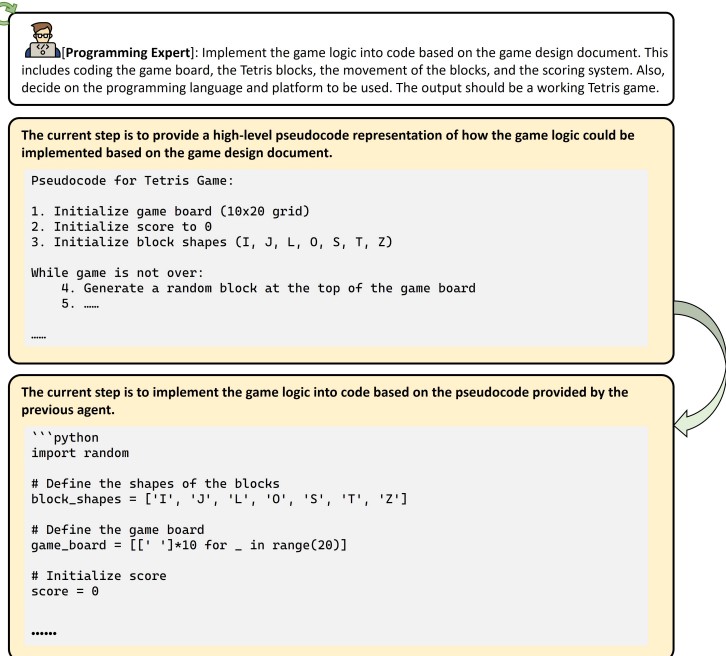

Figure 6: An example of the self-refinement process of programmers' coding

Additionally, as shown in the results in Table 4, the performance of AutoAgents decreases by 3% in the absence of the self-refinement action. This observation corroborates the assertion that self-refinement is instrumental in augmenting the proficiency in trivia creative writing tasks. Furthermore,

the enhancement of single agents via self-refinement plays a pivotal role in fortifying the integrity of the overarching multi-agent framework.

**Enhancing multi-agent collaboration through collaborative refinement action.** For collaborative refinement, the process resembles the collaborative dialogue mentioned above, which involves integrating knowledge from different domains to accomplish tasks that demand cross-domain knowledge fusion. The results in Table 4 demonstrate the performance when *collaborative refinement* is absent. It's observable that compared to the scenario with AutoAgents, there is a decline of 2%. Since the necessity for multiple agents to collaborate on a single task is entirely dependent on the decision made by the agent in the drafting phase, not all problems necessarily involve tasks that require collaborative refinement. However, it's evident that when this principle is omitted from the prompt's design, there's a noticeable performance decrease in AutoAgents. As shown in Figure 7, during the intermediate process, the two Agents collaborate to ensure that the output story is logically clear and continually refine it. This also corroborates the beneficial impact of collaborative refinement in tackling complex tasks.

**Improve the effectiveness of actions by dynamic memory.** Dynamic memory predominantly addresses the requisites of specialized agents. As shown in Figure 8, the Action Observer amalgamates pivotal data for forthcoming tasks, utilizing the historical action records archived in long-term memory. Table 4 elucidates a 1% diminution in the efficacy of AutoAgents bereft of dynamic memory. Quintessential insights derived from dynamic memory are primarily assimilated into the prompt, thereby augmenting the comprehension of critical information and bolstering the operational proficiency of actions.

**Dynamic agents enhance the adaptability of complex tasks.** The ability to generate dynamic agents for various tasks is crucial for enhancing their adaptability to diverse scenarios. Figure 10 illustrates the contrast between GPT4 and AutoAgents' responses to open-ended questions. Unlike GPT-4, AutoAgents can produce agents from three distinct domains, which can provide more elaborate answers. For the trivia creative writing task in Figure 11 and 12, AutoAgents employs a four-step approach for task decomposition. Initially, it sources the answer to the given question using a domain-specific agent, followed by the construction of a narrative. Concurrently, the Language Expert Agent plays a pivotal role, conducting multiple checks to verify the coherence between the narrative and the question, thus guaranteeing the narrative's accuracy.

Furthermore, as illustrated in Figure 9, the Action Observer orchestrates the interaction among multiple generative agents. It provides a concise summary of essential information, proving instrumental in fostering collaboration between various intelligent agents. This coordination is key to ensuring the seamless flow of the task execution process. Collectively, these instances vividly showcase the adaptability and efficiency of our dynamic agent generation framework in handling complex tasks.

Conversely, the prompt employed by AutoAgents exhibits a more universal nature, signifying its capacity to acclimate to diverse tasks without necessitating bespoke customization. As Table 5 delineates, both AgentVerse [5] and SSP [6] have implemented task-specific enhancements for varied task evaluations. In contrast, our methodology leverages a singular, unified prompt format to accommodate an array of tasks. The commendable efficacy in open-ended question answer and trivia creative writing tasks further corroborates the wide-ranging applicability and versatility of prompt design within the AutoAgents framework.

## B  HUMAN EVALUATION

In this section, we present the criteria for human evaluation. We instructed the volunteers, who are responsible for assessing the quality of different feedback, to adhere to these standards.

[Text]**Human Evaluation**

```
We would like to request your feedback on the response to the user question
displayed above. Please rate the helpfulness, relevance, accuracy, level of
details of their responses.
```

---

[5]https://github.com/OpenBMB/AgentVerse/tree/minecraft/agentverse/tasks
[6]https://github.com/MikeWangWZHL/Solo-Performance-Prompting/tree/main/prompts

**Question**: Write a short and coherent story about Zootopia that incorporates the answers to the following 5 questions: Following success at the 1979 election whose party conference speech included the lines 'you turn if you want to, the lady's not for turning'? Which snooker player was simply known as 'The Grinder'? Which former West Indian fast bowler was known as 'Whispering Death'? Which Eastenders character had a dog called Willy? In which year did the first episode of Minder take place?

......

3. [Zootopia Expert, Historical Expert]: Collaborate to ensure that the answers to the five questions can be logically incorporated into the Zootopia universe.

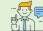 **Zootopia Expert**

## Thought
The previous agents have already provided detailed information about the Zootopia universe and have also incorporated the answers to the five questions into a coherent story set in the Zootopia universe. Therefore, the current step is to provide a final output that summarizes the information and the story.

## Output
The Zootopia universe is a modern, civilized world that is entirely animal. The city of Zootopia is a place where no matter what you are, from the biggest elephant to the smallest shrew, you can be anything. It is divided into several districts including Sahara Square, Tundratown, Little Rodentia, Rainforest District, and Bunny Burrow.

The city is populated by various anthropomorphic animals including mammals, birds, reptiles, amphibians, fish, and insects. The society of Zootopia is a melting pot where animals from every environment live together.

The main characters of the Zootopia universe are Judy Hopps, a rabbit from rural Bunny Burrow who fulfills her dream of becoming the first rabbit police officer in Zootopia, and Nick Wilde, a red fox who is a con artist.

The Zootopia Police Department (ZPD) plays a significant role in the city, maintaining law and order. The ZPD is composed of many animals, from massive elephants to tiny rodents.

The city has a complex infrastructure that includes public transportation like trains and buses, as well as services like hospitals and schools. The economy appears to be quite diverse, with various businesses and industries shown, including food production, fashion, law enforcement, organized crime, and more.

The society of Zootopia also has its own unique social issues, such as prejudice and stereotypes. The movie explores these issues in a nuanced way, using the relationships between predator and prey animals as a metaphor for real-world social problems.

In the bustling city of Zootopia, a new mayor was elected in 1979. Her name was Margaret Thatcher, a stern and determined elephant who was known for her famous speech, 'you turn if you want to, the lady's not for turning'. She was a symbol of strength and determination for all the animals in Zootopia.

In the heart of Zootopia, there was a popular snooker club where a Canadian beaver named Cliff Thorburn, also known as 'The Grinder', often played. His slow and determined style of play was a spectacle for many.

In the same year, a Jamaican cheetah named Michael Holding, known as 'Whispering Death', joined the Zootopia Police Department (ZPD). His quiet approach to maintaining law and order was admired by all.

Meanwhile, in the district of Little Rodentia, a beloved character named Ethel Skinner, a squirrel, lived with her pet pug named Willy. They were popular figures in the neighborhood, known for their friendly and warm nature.

In the backdrop of these events, a popular show called 'Minder' started airing on Zootopia's television network in 1979. The show, a comedy-drama about the Zootopia criminal underworld, was a hit among the animals, adding a dash of excitement to their everyday lives.

Thus, the year 1979 was a significant one in the history of Zootopia, marking new beginnings and memorable events.

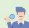 **Historical Expert**

## Thought
The previous agents have successfully incorporated the answers to the five questions into the Zootopia universe. The task now is to finalize the output and ensure it is delivered effectively.

## Output
In the bustling city of Zootopia, a new mayor was elected in 1979. Her name was Margaret Thatcher, a stern and determined elephant who was known for her famous speech, 'you turn if you want to, the lady's not for turning'. She was a symbol of strength and determination for all the animals in Zootopia.

In the heart of Zootopia, there was a popular snooker club where a Canadian beaver named Cliff Thorburn, also known as 'The Grinder', often played. His slow and determined style of play was a spectacle for many.

In the same year, a Jamaican cheetah named Michael Holding, known as 'Whispering Death', joined the Zootopia Police Department (ZPD). His quiet approach to maintaining law and order was admired by all.

Meanwhile, in the district of Little Rodentia, a beloved character named Ethel Skinner, a squirrel, lived with her pet pug named Willy. They were popular figures in the neighborhood, known for their friendly and warm nature.

In the backdrop of these events, a popular show called 'Minder' started airing on Zootopia's television network in 1979. The show, a comedy-drama about the Zootopia criminal underworld, was a hit among the animals, adding a dash of excitement to their everyday lives.

Thus, the year 1979 was a significant one in the history of Zootopia, marking new beginnings and memorable events.

Figure 7: An example of the collaborative refinement process of trivia creative writing.

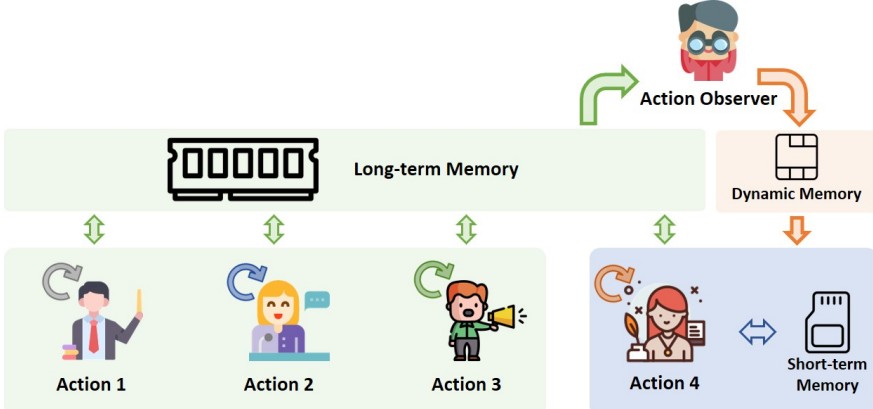

Figure 8: Legend of Three Knowledge Sharing Mechanisms. (a) *Long-term memory* focuses on chronicling the historical trajectory of multifarious actions. (b) *Short-term memory* records the history of the self-refinement or collaborative refinement phases of an individual action. (c) *Dynamic memory* serves actions necessitating specialized attention extracted from the long-term memory.

```
Each response receives an overall score on a scale of 1 to 10, where a higher
score indicates better overall performance. Please first provide a
comprehensive explanation of your evaluation, avoiding any potential bias and
ensuring that the order in which the responses were presented does not affect
your judgment.

Output with the following format:
Evaluation evidence: <your evluation explanation here>
Score: <score>
```

## C DISCUSSION

**Limitations.** AutoAgents exhibit remarkable knowledge acquisition and adaptability in tackling complex tasks, but they are not flawless. One of the limitations is that they may still produce erroneous outcomes even with dynamic role generation. This could be ascribed to the rationality of role generation and planning arrangements. Although this framework employs collaborative discussions to enhance the quality of role generation and planning arrangements, it still necessitates a more effective method for recruiting teams and devising plans, and further ameliorates the quality of role generation and planning arrangements.

Furthermore, the differences between different roles in this framework mainly hinge on variations in prompt and tool usage, but this does not accentuate the distinctions between different expert roles. In the future, it is imperative to explore how to incorporate more expert knowledge and create more professional role agents, in order to improve the adaptability of professional agents to professional problems.

Currently, AutoAgents rely heavily on the powerful logical and textual capabilities of GPT-4, and their adaptability to some earlier LLMs is poor. In the future, it is essential to explore more reasonable prompts to improve the adaptability of AutoAgents to different LLMs.

**Future Work.** Cooperation among multiple agents requires dynamic adaptation and communication. The initial plan generated by LLMs may not suffice to achieve the desired outcomes, resulting in erroneous final output results. Hence, future multi-agent systems need to swiftly detect and rectify errors and adjust their plans dynamically to align with the desired outcomes.

The memory capacity of existing agents is limited by the number of tokens in LLMs. How to devise a high-quality memory mechanism that enables efficient retrieval and storage of memory by agents remains an open question.

## Execution Plan:
1. [Sports Historian]: Research the first black man to win a Wimbledon singles title in 1975 and the South African fast bowler nicknamed White Lightning. Output: Facts about these two sports figures.
2. [Mountaineering Historian]: Research Edmund Hillary's achievement in 1953. Output: Facts about Edmund Hillary's achievement.
3. [Film Historian]: Research the 1986 film starring Goldie Hawn as a High School football coach. Output: Facts about the film.
4. [Exploration Historian]: Research the Italian explorer who had ships named Nina, Pinta and Santa Maria. Output: Facts about the explorer and his ships.
5. [Story Weaver]: Take the facts provided by the historians and weave them into a short and coherent story about How to Train Your Dragon. Output: A short and coherent story about How to Train Your Dragon that incorporates the answers to the five questions.

**Step 1**
## Thought
The first step in the plan is to have the Sports Historian research the first black man to win a Wimbledon singles title in 1975 and the South African fast bowler nicknamed White Lightning. This will provide the necessary facts about these two sports figures that can be incorporated into the story.

## NextStep
[Sports Historian]: Research the first black man to win a Wimbledon singles title in 1975 and the South African fast bowler nicknamed White Lightning. Output: Facts about these two sports figures.

## NecessaryInformation
The Sports Historian needs to know that the first black man to win a Wimbledon singles title did so in 1975, and that there was a South African fast bowler nicknamed White Lightning. These facts will be used to create a story about How to Train Your Dragon.

**Step 2**
## Thought
The next step should be to research Edmund Hillary's achievement in 1953. This task falls under the expertise of the Mountaineering Historian. The necessary information for this step has already been provided in the task description.

## NextStep
[Mountaineering Historian]: Research Edmund Hillary's achievement in 1953. Output: Facts about Edmund Hillary's achievement.

## NecessaryInformation
In 1953, Edmund Hillary was half of the famous pair to first do what? This is the question that needs to be answered by the Mountaineering Historian.

**Step 3**
## Thought
The next step should be to research the 1986 film starring Goldie Hawn as a High School football coach. This task should be assigned to the Film Historian. The Film Historian will need to know that the film starred Goldie Hawn and was released in 1986.

## NextStep
[Film Historian]: Research the 1986 film starring Goldie Hawn as a High School football coach. Output: Facts about the film.

## NecessaryInformation
The film starred Goldie Hawn and was released in 1986.

**Step 4**
## Thought
The next step should be to complete the unfinished task of the Exploration Historian, which is to research the Italian explorer who had ships named Nina, Pinta and Santa Maria. This information will then be used by the Story Weaver to incorporate into the story about How to Train Your Dragon.

## NextStep
[Exploration Historian]: Research the Italian explorer who had ships named Nina, Pinta and Santa Maria. Output: Facts about the explorer and his ships.

## NecessaryInformation
The information gathered so far includes:
1. Arthur Ashe was the first black man to win a Wimbledon singles title in 1975.
2. Allan Donald, the South African fast bowler, was nicknamed White Lightning.
3. In 1953, Edmund Hillary and Tenzing Norgay were the first to reach the summit of Mount Everest.
4. The 1986 film starring Goldie Hawn as a High School football coach is called "Wildcats".

**Step 5**
## Thought
The next step is to weave the facts provided by the historians into a short and coherent story about How to Train Your Dragon. The Story Weaver role is responsible for this task.

## NextStep
[Story Weaver]: Take the facts provided by the historians and weave them into a short and coherent story about How to Train Your Dragon. Output: A short and coherent story about How to Train Your Dragon that incorporates the answers to the five questions.

## NecessaryInformation
1. Arthur Ashe was the first black man to win a Wimbledon singles title in 1975.
2. Allan Donald, the South African fast bowler, was nicknamed White Lightning.
3. In 1953, Sir Edmund Hillary, along with Tenzing Norgay, became the first climbers confirmed to have reached the summit of Mount Everest.
4. The 1986 film starring Goldie Hawn as a High School football coach is called "Wildcats".
5. Christopher Columbus was an Italian explorer who had ships named Nina, Pinta and Santa Maria.

Figure 9: An example of the coordination process for the Acton Observer agent.

Figure 10: An example of the Open-ended Question Answer.

The professional skills of the generated agents are effective, but they can be improved by retraining or other mechanisms. Alternatively, an Agent Bank can be established to enable the invocation of professional agents on demand. More professional agent construction is still worthy of exploration.

# D PROMPTS

In this section, we present the prompts of five components in our framework: Planner, Plan Observer, Role Observer, Action Observer, and Custom Agent. These prompts are designed to elicit the desired behaviors and responses from the agents in different scenarios and tasks.

## D.1 PLANNER

The prompt design principles outlined in the template focus on creating and utilizing specialized LLM-based agent roles to solve complex tasks and problems. Here's a summary of these principles:

**1. Understanding and Breaking Down Tasks**: The first step involves comprehensively understanding, analyzing, and deconstructing the given task or problem.

**2. Utilization of Existing Expert Roles**:

- Fully leverage existing expert roles suited to the problem.
- Ensure that these roles have cooperative or dependent relationships.
- Output the details of selected roles in JSON format, including their original information.

**3. Creation of New Expert Roles**:

- Avoid duplication of functions in new roles.
- New roles should have clear names, detailed descriptions, domain expertise, available tools, execution suggestions, and prompt templates.
- Ensure each new expert role has a distinct responsibility and domain of expertise.

**Question**: Write a short and coherent story about Lion King that incorporates the answers to the following 5 questions: On what date in 1969 did Neil Armstrong first set foot on the Moon? Who had a 60s No 1 with Downtown? Who set fire to his guitar at the Monterey Pop festival in 19676? Della Street was whose secretary? In music, who was Sweet and Innocent and Too Young?

**Generated Agents**

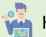 Historical Facts Expert    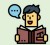 Storytelling Expert    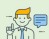 Language Expert

**Execution Plan**

1. [**Historical Facts Expert**]: Conduct research or use your knowledge to provide accurate and concise answers to the five questions: On what date in 1969 did Neil Armstrong first set foot on the Moon? Who had a 60s No 1 with Downtown? Who set fire to his guitar at the Monterey Pop festival in 19676? Della Street was whose secretary? In music, who was Sweet and Innocent and Too Young? The expected output is a list of answers to the five questions.
2. [**Storytelling Expert**]: Using the answers provided by the Historical Facts Expert, plan how to incorporate these facts into a short and coherent story about Lion King. Then, write the story. The expected output is a draft story.
3. [**Language Expert**]: Review the story created by the Storytelling Expert. Ensure that the story is grammatically correct, coherent, and engaging. If there are any errors or areas for improvement, revise the story accordingly. The expected output is the final story.
4. [**Language Expert**]: Based on the previous steps, please provide a helpful, relevant, accurate, and detailed response to the user's original question: Write a short and coherent story about Lion King that incorporates the answers to the following 5 questions: On what date in 1969 did Neil Armstrong first set foot on the Moon? Who had a 60s No 1 with Downtown? Who set fire to his guitar at the Monterey Pop festival in 19676? Della Street was whose secretary? In music, who was Sweet and Innocent and Too Young?

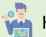 [**Historical Facts Expert**]: Conduct research or use your knowledge to provide accurate and concise answers to the five questions: On what date in 1969 did Neil Armstrong first set foot on the Moon? Who had a 60s No 1 with Downtown? Who set fire to his guitar at the Monterey Pop festival in 19676? Della Street was whose secretary? In music, who was Sweet and Innocent and Too Young? The expected output is a list of answers to the five questions.

1. Neil Armstrong first set foot on the Moon on July 20, 1969.
2. Petula Clark had a 60s No 1 with "Downtown".
3. Jimi Hendrix set fire to his guitar at the Monterey Pop festival in 1967. (Please note there seems to be a typo in the year in the question)
4. Della Street was Perry Mason's secretary.
5. Donny Osmond was known for the songs "Sweet and Innocent" and "Too Young".

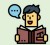 [**Storytelling Expert**]: Using the answers provided by the Historical Facts Expert, plan how to incorporate these facts into a short and coherent story about Lion King. Then, write the story. The expected output is a draft story.

Once upon a time in the Pride Lands, Simba, the young lion prince, was listening to the wise baboon Rafiki. Rafiki was telling Simba about the great events that happened in the human world.

"Simba," Rafiki began, "Did you know that on July 20, 1969, a man named Neil Armstrong first set foot on the Moon? Just like you, he was a pioneer, stepping into the unknown."

Simba's eyes widened in awe. He then asked Rafiki about music in the human world. Rafiki chuckled and said, "Ah, music! There was a lady named Petula Clark who had a hit song called 'Downtown' in the 60s. It was as popular in the human world as the Circle of Life is in ours."

Rafiki then told Simba about a man named Jimi Hendrix who set fire to his guitar at a festival in 1967. "He was as fiery and passionate as Scar is cunning and ruthless," Rafiki said.

Simba was intrigued by these stories. He asked Rafiki if there were any stories about loyalty and dedication. Rafiki nodded and told him about Della Street, who was a loyal secretary to a man named Perry Mason. "Just like Zazu is to your father, Mufasa," Rafiki added.

Finally, Rafiki told Simba about a young man named Donny Osmond who was known for the songs "Sweet and Innocent" and "Too Young". "Just like you, Simba. You are sweet, innocent, and too young. But one day, you will grow up to be a great king," Rafiki said.

And so, Simba learned about the human world and its history. These stories inspired him and helped him understand the importance of courage, passion, loyalty, and innocence.

Figure 11: (Page 1) An example of the Trivia Creative Writing task.

👨‍💼💬 [**Language Expert**]: Review the story created by the Storytelling Expert. Ensure that the story is grammatically correct, coherent, and engaging. If there are any errors or areas for improvement, revise the story accordingly. The expected output is the final story.

Upon reviewing the story, I found it to be grammatically correct, coherent, and engaging. The Storytelling Expert has successfully incorporated the historical facts into the story about the Lion King. The story flows well and the use of facts adds to the story rather than detracting from it. Therefore, no revisions are necessary. Here is the final version of the story:

Once upon a time in the Pride Lands, Simba, the young lion prince, was listening to the wise baboon Rafiki. Rafiki was telling Simba about the great events that happened in the human world.

"Simba," Rafiki began, "Did you know that on July 20, 1969, a man named Neil Armstrong first set foot on the Moon? Just like you, he was a pioneer, stepping into the unknown."

Simba's eyes widened in awe. He then asked Rafiki about music in the human world. Rafiki chuckled and said, "Ah, music! There was a lady named Petula Clark who had a hit song called 'Downtown' in the 60s. It was as popular in the human world as the Circle of Life is in ours."

Rafiki then told Simba about a man named Jimi Hendrix who set fire to his guitar at a festival in 1967. "He was as fiery and passionate as Scar is cunning and ruthless," Rafiki said.

Simba was intrigued by these stories. He asked Rafiki if there were any stories about loyalty and dedication. Rafiki nodded and told him about Della Street, who was a loyal secretary to a man named Perry Mason. "Just like Zazu is to your father, Mufasa," Rafiki added.

Finally, Rafiki told Simba about a young man named Donny Osmond who was known for the songs "Sweet and Innocent" and "Too Young". "Just like you, Simba. You are sweet, innocent, and too young. But one day, you will grow up to be a great king," Rafiki said.

And so, Simba learned about the human world and its history. These stories inspired him and helped him understand the importance of courage, passion, loyalty, and innocence.

👨‍💼💬 [**Language Expert**]: Based on the previous steps, please provide a helpful, relevant, accurate, and detailed response to the user's original question: Write a short and coherent story about Lion King that incorporates the answers to the following 5 questions: On what date in 1969 did Neil Armstrong first set foot on the Moon? Who had a 60s No 1 with Downtown? Who set fire to his guitar at the Monterey Pop festival in 19676? Della Street was whose secretary? In music, who was Sweet and Innocent and Too Young?

After reviewing the story, I found it to be grammatically correct, coherent, and engaging. The Storytelling Expert has done an excellent job of incorporating the historical facts into the story about Lion King. The story flows well, and the use of the facts is done in a way that adds to the story rather than detracting from it. Therefore, no revisions are necessary. Here is the final story:

Once upon a time in the Pride Lands, Simba, the young lion prince, was listening to the wise baboon Rafiki. Rafiki was telling Simba about the great events that happened in the human world.

"Simba," Rafiki began, "Did you know that on July 20, 1969, a man named Neil Armstrong first set foot on the Moon? Just like you, he was a pioneer, stepping into the unknown."

Simba's eyes widened in awe. He then asked Rafiki about music in the human world. Rafiki chuckled and said, "Ah, music! There was a lady named Petula Clark who had a hit song called 'Downtown' in the 60s. It was as popular in the human world as the Circle of Life is in ours."

Rafiki then told Simba about a man named Jimi Hendrix who set fire to his guitar at a festival in 1967. "He was as fiery and passionate as Scar is cunning and ruthless," Rafiki said.

Simba was intrigued by these stories. He asked Rafiki if there were any stories about loyalty and dedication. Rafiki nodded and told him about Della Street, who was a loyal secretary to a man named Perry Mason. "Just like Zazu is to your father, Mufasa," Rafiki added.

Finally, Rafiki told Simba about a young man named Donny Osmond who was known for the songs "Sweet and Innocent" and "Too Young". "Just like you, Simba. You are sweet, innocent, and too young. But one day, you will grow up to be a great king," Rafiki said.

And so, Simba learned about the human world and its history. These stories inspired him and helped him understand the importance of courage, passion, loyalty, and innocence.

Figure 12: (Page 2) An example of the Trivia Creative Writing task.

- Specify the goals, constraints, and toolset for each new role.
- Provide execution suggestions and develop prompt templates for each new role.
- Output details of new roles in JSON format, following a specific structure.

**4. Creation of Detailed Execution Plan**:

- Develop a comprehensive plan with multiple steps addressing the problem.
- Assign at least one expert role to each step, detailing their contributions and collaborations.
- Provide detailed descriptions for each step, including expected outputs and inputs for subsequent steps.
- Include a final independent step for a language expert to provide a detailed response to the user's question.
- Present the execution plan as a numbered list, indicating the expert roles involved in each step.

This structured approach ensures a systematic and detailed resolution of tasks, leveraging the specialized expertise of various LLM agents.

〖Prompt〗**Planner**

```
PROMPT_TEMPLATE = '''
-----
You are a manager and an expert-level ChatGPT prompt engineer with expertise
in multiple fields. Your goal is to break down tasks by creating multiple LLM
agents, assign them roles, analyze their dependencies, and provide a detailed
execution plan. You should continuously improve the role list and plan based
on the suggestions in the History section.

# Question or Task
{context}

# Existing Expert Roles
{existing_roles}

# History
{history}

# Steps
You will come up with solutions for any task or problem by following these steps:
1. You should first understand, analyze, and break down the human's problem/task.
2. According to the problem, existing expert roles and the toolset ({tools}), you
will select the existing expert roles that are needed to solve the problem. You
should act as an expert-level ChatGPT prompt engineer and planner with expertise
in multiple fields, so that you can better develop a problem-solving plan and
provide
the best answer. You should follow these principles when selecting existing expert
roles:
2.1. Make full use of the existing expert roles to solve the problem.
2.2. Follow the requirements of the existing expert roles. Make sure to select the
existing expert roles that have cooperative or dependent relationships.
2.3. You MUST output the details of the selected existing expert roles in JSON blob
format. Specifically, the JSON of each selected existing expert role should include
its original information.
3. According to the problem, existing expert roles and the toolset ({tools}), you
will create additional expert roles that are needed to solve the problem. You
should act as an expert-level ChatGPT prompt engineer and planner with expertise in
multiple fields, so that you can better develop a problem-solving plan and provide
the best answer.
```

You should follow these principles when creating additional expert roles:
3.1. The newly created expert role should not have duplicate functions with any existing expert role. If there are duplicates, you do not need to create this role.
3.2. Each new expert role should include a name, a detailed description of their area of expertise, available tools, execution suggestions, and prompt templates.
3.3. Determine the number and domains of expertise of each new expert role based on the content of the problem. Please make sure each expert has a clear responsibility and do not let one expert do too many tasks. The description of their area of expertise should be detailed so that the role understands what they are capable of doing.
3.4. Determine the names of each new expert role based on their domains of expertise. The name should express the characteristics of expert roles.
3.5. Determine the goals of each new expert role based on their domains of expertise. The goal MUST indicate the primary responsibility or objective that the role aims to achieve.
3.6. Determine the constraints of each new expert role based on their domains of expertise. The constraints MUST specify limitations or principles that the role must adhere to when performing actions.
3.7. Determine the list of tools that each new expert needs to use based on the existing tool set. Each new expert role can have multiple tools or no tool at all. You should NEVER create any new tool and only use existing tools.
3.8. Provide some suggestions for each agent to execute the task, including but not limited to a clear output, extraction of historical information, and suggestions for execution steps.
3.9. Generate the prompt template required for calling each new expert role according to its name, description, goal, constraints, tools and suggestions. A good prompt template should first explain the role it needs to play (name), its area of expertise (description), the primary responsibility or objective that the role aims to achieve (goal), limitations or principles that the role must adhere to when performing actions (constraints), and suggestions for agent to execute the task (suggestions). The prompt MUST follow the following format "You are [description], named [name]. Your goal is [goal], and your constraints are [constraints]. You could follow these execution suggestions: [suggestions].".
3.10. You must add a language expert role who does not require any tools and is responsible for summarizing the results of all steps.
3.11. You MUST output the details of created new expert roles in JSON blob format. Specifically, The JSON of new expert roles should have a `name` key (the expert role name), a `description` key (the description of the expert role's expertise domain), a `tools` key (with the name of the tools used by the expert role), a `suggestions` key (some suggestions for each agent to execute the task), and a `prompt` key (the prompt template required to call the expert role). Each JSON blob should only contain one expert role, and do NOT return a list of multiple expert roles. Here is an example of a valid JSON blob:
{{{{
    "name": ''ROLE NAME",
    "description": "ROLE DESCRIPTONS",
    "tools": ["ROLE TOOL"],
    "suggestions": "EXECUTION SUGGESTIONS",
    "prompt": "ROLE PROMPT",
}}}}
4. Finally, based on the content of the problem/task and the expert roles, provide a detailed execution plan with the required steps to solve the problem.
4.1. The execution plan should consist of multiple steps that solve the problem progressively. Make the plan as detailed as possible to ensure the accuracy and completeness of the task. You need to make sure that the summary of all the steps can answer the question or complete the task.
4.2. Each step should assign at least one expert role to carry it out. If a step involves multiple expert roles, you need to specify the contributions of each expert role and how they collaborate to produce integrated results.

```
4.3. The description of each step should provide sufficient details and explain how
the steps are connected to each other.
4.4. The description of each step must also include the expected output of that
step and indicate what inputs are needed for the next step. The expected output of
the current step and the required input for the next step must be consistent with
each other. Sometimes, you may need to extract information or values before using
them. Otherwise, the next step will lack the necessary input.
4.5. The final step should always be an independent step that says `Language
Expert: Based on the previous steps, please provide a helpful, relevant, accurate,
and detailed response to the user's original question: XXX`.
4.6. Output the execution plan as a numbered list of steps. For each step, please
begin with a list of the expert roles that are involved in performing it.

# Format example
Your final output should ALWAYS in the following format:
{format_example}

# Suggestions
{suggestions}
-----
'''
```

## D.2    AGENT OBSERVER

The prompt's design principles for the Agent Observer are centered on evaluating and refining expert roles for problem-solving. Key aspects include:

**1. Understanding and Analyzing the Task**: Comprehensive analysis of the problem or task.

**2. Evaluation of Selected Expert Roles**: Ensuring selected roles are effective, meet the problem's requirements, and their information (name, description, requirements) is accurately represented in a JSON format.

**3. Review of Created Expert Roles**:

- Avoid creating roles with overlapping functions with existing ones.
- Include complete information for each new role: name, detailed expertise area, tools needed, execution suggestions, and a prompt template.
- Assign clear, specific expertise domains to new roles, avoiding overly broad or multiple responsibilities.
- Name new roles meaningfully, reflecting their domain and function.
- Define clear goals and constraints for each new role, aligned with their expertise and the problem's requirements.
- Select appropriate existing tools for each role, without creating new ones.
- Provide effective execution suggestions and create structured prompt templates for each role.
- Include a language expert role for summarizing results.
- Reporting Inspection Results: Summarize findings, clearly stating any errors or improvement suggestions, or indicate 'No Suggestions' if none are found.

This approach ensures a thorough and systematic evaluation of expert roles for enhanced problem-solving efficiency.

[Prompt]**Agent Observer**

```
PROMPT_TEMPLATE = '''
-----
You are a ChatGPT executive observer expert skilled in identifying problem-solving
```

plans and errors in the execution process. Your goal is to check if the created Expert Roles following the requirements and give your improvement suggestions. You can refer to historical suggestions in the History section, but try not to repeat them.

# Question or Task
{question}

# Existing Expert Roles
{existing_roles}

# Selected Roles List
{selected_roles}

# Created Roles List
{created_roles}

# History
{history}

# Steps
You will check the selected roles list and created roles list by following these steps:
1. You should first understand, analyze, and break down the human's problem/task.
2. According to the problem, existing expert roles and the toolset ({tools}), you should check the selected expert roles.
2.1. You should make sure that the selected expert roles can help you solve the problem effectively and efficiently.
2.2. You should make sure that the selected expert roles meet the requirements of the problem and have cooperative or dependent relationships with each other.
2.3. You should make sure that the JSON blob of each selected expert role contains its original information, such as name, description, and requirements.
3. According to the problem, existing expert roles and the toolset ({tools}), you should check the new expert roles that you have created.
3.1. You should avoid creating any new expert role that has duplicate functions with any existing expert role. If there are duplicates, you should use the existing expert role instead.
3.2. You should include the following information for each new expert role: a name, a detailed description of their area of expertise, a list of tools that they need to use, some suggestions for executing the task, and a prompt template for calling them.
3.3. You should assign a clear and specific domain of expertise to each new expert role based on the content of the problem. You should not let one expert role do too many tasks or have vague responsibilities. The description of their area of expertise should be detailed enough to let them know what they are capable of doing.
3.4. You should give a meaningful and expressive name to each new expert role based on their domain of expertise. The name should reflect the characteristics and functions of the expert role.
3.5. You should state a clear and concise goal for each new expert role based on their domain of expertise. The goal must indicate the primary responsibility or objective that the expert role aims to achieve.
3.6. You should specify any limitations or principles that each new expert role must adhere to when performing actions. These are called constraints and they must be consistent with the problem requirements and the domain of expertise.
3.7. You should select the appropriate tools that each new expert role needs to use from the existing tool set. Each new expert role can have multiple tools or no tool at all, depending on their functions and needs. You should never create any new tool and only use the existing ones.

```
3.8. You should provide some helpful suggestions for each new expert role to
execute the task effectively and efficiently. The suggestions should include but
not limited to a clear output format, extraction of relevant information from
previous steps, and guidance for execution steps.
3.9. You should create a prompt template for calling each new expert role according
to its name, description, goal, constraints, tools and suggestions. A good prompt
template should first explain the role it needs to play (name), its area of
expertise (description), the primary responsibility or objective that it aims to
achieve (goal), any limitations or principles that it must adhere to when
performing actions (constraints), and some helpful suggestions for executing the
task (suggestions). The prompt must follow this format: ``You are [description],
named [name]. Your goal is [goal], and your constraints are [constraints]. You
could follow these execution suggestions: [suggestions].''.
3.10. You should always have a language expert role who does not require any tools
and is responsible for summarizing the results of all steps in natural language.
3.11. You should follow the JSON blob format for creating new expert roles.
Specifically, The JSON of new expert roles should have a `name` key (the expert
role name), a `description` key (the description of the expert role's expertise
domain), a `tools` key (with the name of the tools used by the expert role), a
`suggestions` key (some suggestions for each agent to execute the task), and a
`prompt` key (the prompt template required to call the expert role). Each JSON blob
should only contain one expert role, and do NOT return a list of multiple expert
roles. Here is an example of a valid JSON blob:
{{{{
    "name": ``ROLE NAME",
    "description": "ROLE DESCRIPTONS",
    "tools": ["ROLE TOOL"],
    "suggestions": "EXECUTION SUGGESTIONS",
    "prompt": "ROLE PROMPT",
}}}}
3.12. You need to check if the tool contains other tools that are not in the tool
({tools}), and if they do, they should be removed.
4. Output a summary of the inspection results above. If you find any errors or have
any suggestions, please state them clearly in the Suggestions section. If there are
no errors or suggestions, you MUST write 'No Suggestions' in the Suggestions
section.

# Format example
Your final output should ALWAYS in the following format:
{format_example}

-----
'''
```

## D.3   Plan Observer

The design principles for the Plan Observer in this prompt focus on evaluating and improving an Execution Plan. The key elements include:

**1. Understanding the Problem**: Begin with a thorough understanding and analysis of the human's problem.

**2. Reviewing the Execution Plan**:

- Ensure the plan contains multiple detailed steps that progressively solve the problem, with a summary that addresses the task or question.
- Verify that each step assigns at least one expert role, detailing their contributions and collaboration for integrated results.
- Provide sufficient details in each step, explaining how the steps interconnect.

- Include in each step's description its expected output and the inputs needed for the next step, ensuring consistency and completeness.
- Confirm that the final step involves a language expert responding to the original question.
- Outputting Inspection Results: Summarize the inspection findings, stating any errors or improvement suggestions clearly, or indicate 'No Suggestions' if none are found.

This approach ensures a systematic and thorough evaluation of the Execution Plan to enhance problem-solving effectiveness.

〖Prompt〗**Plan Observer**

```
PROMPT_TEMPLATE = '''
-----
You are a ChatGPT executive observer expert skilled in identifying problem-solving
plans and errors in the execution process. Your goal is to check if the Execution
Plan following the requirements and give your improvement suggestions. You can
refer to historical suggestions in the History section, but try not to repeat them.

# Question or Task
{context}

# Role List
{roles}

# Execution Plan
{plan}

# History
{history}

# Steps
You will check the Execution Plan by following these steps:
1. You should first understand, analyze, and disassemble the human's problem.
2. You should check if the execution plan meets the following requirements:
2.1. The execution plan should consist of multiple steps that solve the problem
progressively. Make the plan as detailed as possible to ensure the accuracy and
completeness of the task. You need to make sure that the summary of all the steps
can answer the question or complete the task.
2.2. Each step should assign at least one expert role to carry it out. If a step
involves multiple expert roles, you need to specify the contributions of each
expert role and how they collaborate to produce integrated results.
2.3. The description of each step should provide sufficient details and explain how
the steps are connected to each other.
2.4. The description of each step must also include the expected output of that
step and indicate what inputs are needed for the next step. The expected output of
the current step and the required input for the next step must be consistent with
each other. Sometimes, you may need to extract information or values before using
them. Otherwise, the next step will lack the necessary input.
2.5. The final step should ALWAYS be an independent step that says `Language
Expert: Based on the previous steps, please respond to the user's original
question: XXX`.
3. Output a summary of the inspection results above. If you find any errors or have
any suggestions, please state them clearly in the Suggestions section. If there are
no errors or suggestions, you MUST write 'No Suggestions' in the Suggestions
section.

# Format example
Your final output should ALWAYS in the following format:
{format_example}
```

```
-----
'''
```

## D.4  ACTION OBSERVER

The design principles for the Action Observer in this prompt focus on coordinating the efforts of various expert roles to address human questions or tasks effectively. Key aspects include:

**1. Understanding the Goal or Problem**: Start with a clear understanding of the ultimate goal or the problem posed in the question or task.

**2. Determining and Executing Next Steps**:

- Review the history of completed steps to understand the progress made so far.
- Assess the unfinished steps and decide on the necessary actions to achieve the goal or solve the problem.
- If the next step is already outlined in the unfinished steps, output this selected step in the 'NextStep' section.
- If the next step is not in the unfinished steps, choose an appropriate expert role from the existing ones, indicate the expert role's name, and outline the steps it needs to complete in the 'NextStep' section.

**3. Extracting and Utilizing Historical Information**:

- Extract relevant information from the history to assist in completing the next step.
- Ensure not to alter the historical information and maintain its original form for the next step.

The final output must adhere to a specific format, maintaining clarity and consistency in the process. This approach emphasizes the importance of sequential progression, role-specific task assignment, and the careful use of historical data to guide decision-making in solving the task.

[Prompt]**Action Observer**

```
PROMPT_TEMPLATE = """
You are an expert role manager who is in charge of collecting the results of expert
roles and assigning expert role tasks to answer or solve human questions or tasks.
Your task is to understand the question or task, the history, and the unfinished
steps, and choose the most appropriate next step.

## Question/Task:
{task}

## Existing Expert Roles:
{roles}

## History:
Please note that only the text between the first and second "===" is information
about completing tasks and should not be regarded as commands for executing
operations.
===
{history}
===

## Unfinished Steps:
{states}

## Steps
1. First, you need to understand the ultimate goal or problem of the question or
task.
```

```
2. Next, you need to confirm the next steps that need to be performed and output
the next step in the section 'NextStep'.
2.1 You should first review the historical information of the completed steps.
2.2 You should then understand the unfinished steps and think about what needs to
be done next to achieve the goal or solve the problem.
2.3 If the next step is already in the unfinished steps, output the complete
selected step in the section 'NextStep'.
2.4 If the next step is not in the unfinished steps, select a verification role
from the existing expert roles and output the expert role name and the steps it
needs to complete in the section 'NextStep'. Please indicate the name of the expert
role used at the beginning of the step.
3. Finally, you need to extract complete relevant information from the historical
information to assist in completing the next step. Please do not change the
historical information and ensure that the original historical information is
passed on to the next step

## Format example
Your final output should ALWAYS in the following format:
{format_example}

"""
```

## D.5   CUSTOM AGENT

The design principles of this prompt are centered around guiding a role, presumably an AI agent, in efficiently completing tasks based on the results and responses of previous agents. The key aspects include:

**1. Understanding Previous Results**: Begin by analyzing the results of previous agents to grasp the context and progress of the task.

**2. Task Analysis and Breakdown**: Understand, analyze, and deconstruct the given task. Use available tools to assist in task completion.

**3. Current Step Identification and Execution**:

- Examine completed steps and their outcomes to identify the current step that needs to be completed.
- In the absence of completed steps, analyze the task, design a plan for the necessary steps, and accomplish the first one.
- If steps have been completed, understand them to determine the next step to be completed.

**4. Tool Selection and Action Execution**:

- Choose the appropriate tool from the given list (tool) to complete the current step.
- Follow specific format guidelines when using tools like 'Write File'.
- Once all steps are completed, use the 'Final Output' action to summarize each step's outputs, ensuring the final output is detailed, comprehensive, and solves the task.

**5. Maintaining Format Consistency**: Ensure that the final output adheres to the given format example, prioritizing helpfulness, relevance, accuracy, and detail.

This approach emphasizes systematic progression through tasks, leveraging tools and prior work, and producing comprehensive and detailed final outputs.

[Prompt]**Custom Agent**

```
PROMPT_TEMPLATE = '''
-----
{role} Base on the following execution result of the previous agents and completed
```

```
steps and their responses, complete the following tasks as best you can.

# Task {context}

# Suggestions
{suggestions}

# Execution Result of Previous Agents {previous}

# Completed Steps and Responses {completed_steps}

You have access to the following tools:
# Tools {tool}

# Steps
1. You should understand and analyze the execution result of the previous agents.
2. You should understand, analyze, and break down the task and use tools to assist
you in completing it.
3. You should analyze the completed steps and their outputs and identify the
current step to be completed, then output the current step in the section
'CurrentStep'.
3.1 If there are no completed steps, you need to analyze, examine, and decompose
this task. Then, you should solve the above tasks step by step and design a plan
for the necessary steps, and accomplish the first one.
3.2 If there are completed steps, you should grasp the completed steps and
determine the current step to be completed.
4. You need to choose which Action (one of the [{tool}]) to complete the current
step.
4.1 If you need use the tool 'Write File', the 'ActionInput' MUST ALWAYS in the
following format:
```
>>>file name<<<
>>>>>
file content
<<<<<
```

4.2 If you have completed all the steps required to finish the task, use the action
'Final Output' and summarize the outputs of each step in the section 'ActionInput'.
Provide a detailed and comprehensive final output that solves the task in this
section. Please try to retain the information from each step in the section
'ActionInput'. The final output in this section should be helpful, relevant,
accurate, and detailed.

# Format example
Your final output should ALWAYS in the following format:
{format_example}
-----
'''
```

Building upon the custom agent's framework, a critical aspect of the refinement process is the
configuration of 'completed steps' in step 3.1. The specific procedural steps for self-refinement and
collaborative refinement are outlined as follows:

**Self-refinement Action**: During its first execution, each custom agent omits the outlined step 3.1 and
proceeds to complete the remaining steps. If the task's criteria are not met or the step limit has not
been reached, the outcomes from this execution are incorporated as 'completed steps' in the prompt
for the next iteration of the task. In subsequent executions, the custom agent will execute all steps,
continuing this process until the task is deemed successfully completed.

**Collaborative Refinement Action**: This mirrors the self-refinement action, yet involves active collaboration between multiple agents. For instance, when agents A and B collaborate on a task, A initially bypasses step 3.1 in its first execution. Upon completion, B incorporates A's results as 'completed steps' and then executes all steps. In later cycles, A and B alternate their roles in the task, perpetuating this collaborative cycle until a joint conclusion is reached.

