# OpenReview forum: "AutoAgents: A Framework for Automatic Agent Generation"
_ICLR.cc/2024/Conference — Submitted to ICLR 2024_

### Official Review · Reviewer_ypGp · 2023-10-29

**Soundness:** 2 fair
**Presentation:** 1 poor
**Contribution:** 3 good
**Rating:** 6
**Confidence:** 3

**Summary:**

AutoAgents is a framework for orchestrating multiple specialized
agents dynamically to form AI teams tailored to different tasks. It
divides the process into a Drafting Stage and an Execution Stage,
allowing agents to collaborate effectively.

**Strengths:**

This study provides a valuable clarification of its position,
especially in the context of LLM-based Agent frameworks. In comparison
to AgentVerse and SSP, this research stands out by highlighting the
significance of Self-Refinement agents and Collaborative Refinement
Action as key differentiators.

**Weaknesses:**

The paper is perceived as having low readability and insufficient
reproducibility. The reviewer kindly requests a more granular
description of the methodology that enables readers to implement the
procedures step by step. For instance, while Table 1 is highly
beneficial for positioning this research within the LLM-based Agent
framework, in comparison to AgentVerse and SSP, it distinctly
highlights the significance of Self-Refinement agents and
Collaborative Refinement Action. Nevertheless, the two points
mentioned above are not clearly articulated in Section 3.2, "EXECUTION
STAGE." They are mentioned in the text and Figure 2, but their
presentation as steps is absent, making it challenging for readers to
comprehend and evaluate reproducibility.

The evaluation in the experiments lacks qualitative insights. In the
experiments, it remains unclear how the introduction of
Self-Refinement agents and Collaborative Refinement Action has led to
differential outcomes compared to SSP, and what specific effects these
two points have had. While accuracy has undeniably improved, it is
essential to qualitatively demonstrate how these two aspects have
contributed to the observed results.

There are concerns regarding the reproducibility of the
experiments. It is unclear whether the CASE STUDY has been practically
realized or if it serves as an imagined example for
application.


The paper lacks an ablation study to assess the impact of modifying or
omitting certain components within the system, particularly in the
Draft and Execution phases where multiple agents are involved, such as
Agent Observer, Plan Observer, Researcher, Planner, Writer, Character
Developer, and others. This study could help elucidate the
significance of each component and its contribution to the overall
system.  Furthermore, the absence of an ablation study regarding
Short-term memory, Long-term memory, and Dynamic memory raises
concerns. Investigating the effects of altering or removing these
memory components could provide valuable insights into their
respective roles and importance within the framework.  Overall,
conducting such ablation studies would enhance the paper's depth and
provide a more comprehensive understanding of the system's inner
workings and the role of its individual components.

**Questions:**

Could you provide a more detailed, step-by-step description of the
Self-Refinement agents and Collaborative Refinement Action in Section
3.2, "EXECUTION STAGE," to enhance readability and reproducibility?

Can you offer qualitative insights to elucidate how the introduction
of Self-Refinement agents and Collaborative Refinement Action has
influenced the experiment results, particularly in comparison to SSP,
to help readers better understand the effects of these elements?

Please note that further clarification on the practical realization of
the CASE STUDY would be valuable to address concerns about
reproducibility.

Is there an opportunity for conducting ablation studies to investigate
the importance of individual components in both the Draft and
Execution phases, as well as the Short-term memory, Long-term memory,
and Dynamic memory components in Knowledge Sharing Mechanism? Such
studies could help clarify the significance and roles of these
components in the framework.

These three elements - the number of agents, self-refinement, and
Collaborative Refinement Action - are key characteristics of this
study from Table 1. Can you please explain why the OPEN-ENDED QUESTION
ANSWER and The Trivia Creative Writing tasks are well-suited for
assessing the impact of these factors?

---

> ### Author Response · Authors · 2023-11-19
> **Official Comment by Authors**
>
> Thank you for your motivating review and concrete suggestions. Detailed responses to your questions are listed as follows.
>
> > 1. The paper is perceived as having low readability and insufficient reproducibility. The reviewer kindly requests a more granular description of the methodology that enables readers to implement the procedures step by step. For instance, while Table 1 is highly beneficial for positioning this research within the LLM-based Agent framework, in comparison to AgentVerse and SSP, it distinctly highlights the significance of Self-Refinement agents and Collaborative Refinement Action. Nevertheless, the two points mentioned above are not clearly articulated in Section 3.2, "EXECUTION STAGE." They are mentioned in the text and Figure 2, but their presentation as steps is absent, making it challenging for readers to comprehend and evaluate reproducibility.
> > 2. Could you provide a more detailed, step-by-step description of the Self-Refinement agents and Collaborative Refinement Action in Section 3.2, "EXECUTION STAGE," to enhance readability and reproducibility?
>
> In order to streamline the implementation process of self-refinement and collaborative refinement, we have introduced the concept of a 'custom agent'. The intricate design of its prompt has been meticulously detailed in Appendix D.5. For enhanced clarity and understanding, we have itemized the core design principles of this custom agent below:
>
> 1. Understanding Previous Results: Begin by analyzing the results of previous agents to grasp the context and progress of the task.
>
> 2. Task Analysis and Breakdown: Understand, analyze, and deconstruct the given task. Use available tools to assist in task completion.
>
> 3. Current Step Identification and Execution:
> - 3.1 Examine completed steps and their outcomes to identify the current step that needs to be completed.
> - 3.2 In the absence of completed steps, analyze the task, design a plan for the necessary steps, and accomplish the first one.
> - 3.3 If steps have been completed, understand them to determine the next step to be completed.
>
> 4. Tool Selection and Action Execution:
> - 4.1 Choose the appropriate tool from the given list ({tool}) to complete the current step.
> - 4.2 Follow specific format guidelines when using tools like 'Write File'.
> - 4.3 Once all steps are completed, use the 'Final Output' action to summarize each step's outputs, ensuring the final output is detailed, comprehensive, and solves the task.
>
> 5. Maintaining Format Consistency: Ensure that the final output adheres to the given format example, prioritizing helpfulness, relevance, accuracy, and detail.
>
> Building upon the custom agent's framework, a critical aspect of the refinement process is the configuration of 'completed steps' in step 3.1. The specific procedural steps for self-refinement and collaborative refinement are outlined as follows:
>
> __Self-refinement Action__: During its first execution, each custom agent omits the outlined step 3.1 and proceeds to complete the remaining steps. If the task's criteria are not met or the step limit has not been reached, the outcomes from this execution are incorporated as 'completed steps' in the prompt for the next iteration of the task. In subsequent executions, the custom agent will execute all steps, continuing this process until the task is deemed successfully completed.
>
> __Collaborative Refinement Action__: This mirrors the self-refinement action, yet involves active collaboration between multiple agents. For instance, when agents A and B collaborate on a task, A initially bypasses step 3.1 in its first execution. Upon completion, B incorporates A's results as 'completed steps' and then executes all steps. In later cycles, A and B alternate their roles in the task, perpetuating this collaborative cycle until a joint conclusion is reached.
>
> -----

---

> > ### Author Response · Authors · 2023-11-19
> > **Official Comment by Authors**
> >
> > > The evaluation in the experiments lacks qualitative insights. In the experiments, it remains unclear how the introduction of Self-Refinement agents and Collaborative Refinement Action has led to differential outcomes compared to SSP, and what specific effects these two points have had. While accuracy has undeniably improved, it is essential to qualitatively demonstrate how these two aspects have contributed to the observed results.
> >
> > Within Appendix A of our revision, we have incorporated an ablation study specifically focused on AutoAgents within the context of trivia creative writing task. The table provided below delineates the performance fluctuations of AutoAgents when certain key components are omitted, thereby illustrating the impact of each individual element on the overall efficacy of the system.
> > - __AutoAgents w/o observers__: During the Drafting Stage, the Planner in AutoAgents engages in collaborative discussions with two Observers to determine the optimal list of agents and the execution plan. Table below elucidates that in the absence of observers, there is a marked 3% reduction in the overall performance of AutoAgents. This substantiates the imperative role of collaborative discussions in agent generation.
> > - __AutoAgents w/o self-refinement__: The performance of AutoAgents decreases by 3% in the absence of the self-refinement agent. This observation corroborates the assertion that self-refinement is instrumental in augmenting proficiency in trivia creative writing tasks. Furthermore, the enhancement of single agents via self-refinement plays a pivotal role in fortifying the integrity of the overarching multi-agent framework.
> > - __AutoAgents w/o collaborative refinement__: It's observable that compared to the scenario with AutoAgents, there is a decline of 2% without collaborative refinement. Since the necessity for multiple agents to collaborate on a single task is entirely dependent on the decision made by the agent in the drafting phase, not all problems necessarily involve tasks that require collaborative refinement. However, it's evident that when this principle is omitted from the prompt's design, there's a noticeable performance decrease in AutoAgents. This also corroborates the beneficial impact of collaborative refinement in tackling complex tasks.
> > - __AutoAgents w/o dynamic memory__: Dynamic memory predominantly addresses the requisites of specialized agents. The Action Observer amalgamates pivotal data for forthcoming tasks, utilizing the historical action records archived in long-term memory. The below table elucidates a 1% diminution in the efficacy of AutoAgents bereft of dynamic memory. Quintessential insights derived from dynamic memory are primarily assimilated into the prompt, thereby augmenting the comprehension of critical information and bolstering the operational proficiency of actions.
> >
> > | Method | N (# triva questions) = 5|
> > | --- | --- |
> > | SSP | 84.4 |
> > | AutoAgents w/o observers | 87.0 |
> > | AutoAgents w/o self-refinement | 87.0 |
> > | AutoAgents w/o collaborative refinement | 88.0  |
> > | AutoAgents w/o dynamic memory | 89.0  |
> > | AutoAgents | 90.0 |
> >
> > -----
> >
> > > 1. There are concerns regarding the reproducibility of the experiments. It is unclear whether the CASE STUDY has been practically realized or if it serves as an imagined example for application.
> > > 2. Please note that further clarification on the practical realization of the CASE STUDY would be valuable to address concerns about reproducibility.
> >
> > The CASE STUDY showcased within our paper are authentic execution outcomes from our study. To offer a more comprehensive understanding, we have included a video in the supplementary materials. This video illustrates the dynamic process of AutoAgents generating and executing code, thereby providing a tangible demonstration of our system's capabilities in action.
> >
> > -----

---

> > > ### Author Response · Authors · 2023-11-19
> > > **Official Comment by Authors**
> > >
> > > > The paper lacks an ablation study to assess the impact of modifying or omitting certain components within the system, particularly in the Draft and Execution phases where multiple agents are involved, such as Agent Observer, Plan Observer, Researcher, Planner, Writer, Character Developer, and others. This study could help elucidate the significance of each component and its contribution to the overall system.
> > >
> > > We would like to emphasize that AutoAgents are fundamentally engineered to create unique agents tailored for varied tasks. Within this design, only a select group of agents – the Planner, Agent Observer, Plan Observer, and Action Observer – are pre-established. The Planner and Action Observer, playing pivotal roles in the drafting and execution phases, respectively, are essential components of the framework and are indispensable. Consequently, our ablation study focused explicitly on examining the impact resulting from the exclusion of the Agent Observer and Plan Observer. For a more detailed exploration of these effects, we direct readers to the ablation experiment section in the section previously mentioned.
> > >
> > > -----
> > >
> > > > Furthermore, the absence of an ablation study regarding Short-term memory, Long-term memory, and Dynamic memory raises concerns. Investigating the effects of altering or removing these memory components could provide valuable insights into their respective roles and importance within the framework.
> > >
> > > To further elucidate, we delineate the specific agent types that are compatible with the three modes of knowledge sharing, as depicted in Figure 7 of our revised manuscript:
> > >
> > > - __Short-term Memory__: This is predominantly utilized by agents that emerge from self-refinement and collaborative-refinement processes. These agents leverage short-term memory to retain historical records throughout the refinement phase. Notably, every agent generated within our framework actively employs short-term memory.
> > >
> > > - __Long-term Memory__: This component archives the execution results of each agent, primarily chronicling the cumulative progress of the execution plan. Access to long-term memory is exclusive to a pre-designated Action Observer. The Action Observer utilizes this memory to evaluate whether additional execution steps are necessary or if there's a need to generate dynamic memory content.
> > >
> > > - __Dynamic Memory__: This memory type is essential for distilling and summarizing pivotal information from long-term memory for diverse agents. Generated by the Action Observer, dynamic memory serves to amalgamate critical aspects of the overall execution plan, thereby facilitating the agents responsible for implementing subsequent tasks. Consequently, the dynamic memory content available to each generated agent may differ, contingent upon their specific roles and tasks.
> > >
> > > Due to the fundamental nature of Short-term Memory and Long-term Memory as indispensable components within the system, our investigation was exclusively focused on understanding and analyzing the role of dynamic memory.
> > > In the Appendix A of the revision, we have included an ablation study of dynamic memory. The following table shows the results without the dynamic memory. It is observed that the performance of AutoAgents decreases by 1% without the summarization from dynamic memory, which also proves the importance of dynamic memory.
> > >
> > > | Method | N (# triva questions) = 5|
> > > | --- | --- |
> > > | AutoAgents w/o dynamic memory | 89.0  |
> > > | AutoAgents | 90.0 |
> > >
> > > -----

---

> > > > ### Author Response · Authors · 2023-11-19
> > > > **Official Comment by Authors**
> > > >
> > > > > Can you offer qualitative insights to elucidate how the introduction of Self-Refinement agents and Collaborative Refinement Action has influenced the experiment results, particularly in comparison to SSP, to help readers better understand the effects of these elements?
> > > >
> > > > | Method | N (# triva questions) = 5|
> > > > | --- | --- |
> > > > | SSP | 84.4 |
> > > > | AutoAgents w/o self-refinement | 87.0 |
> > > > | AutoAgents w/o collaborative refinement | 88.0  |
> > > > | AutoAgents | 90.0 |
> > > >
> > > > __Enhancing single-agent through self-refinement__.
> > > > Self-Refinement is a technique that enables LLMs to “converse” with themselves, evaluate their own generation, and iteratively improve their answers. Although AutoAgents is a framework for multi-agent collaboration, it also requires self-refinement agents to perform specialized roles for individual tasks. Figure 6 in Appendix A depicts the self-refinement process of programmers' coding. They first write a pseudo code file and then generate the corresponding program files based on it. This refinement process significantly ensures the validity of the output file. Additionally, as shown in the results in the above table, the performance of AutoAgents decreases by 3% in the absence of the self-refinement action. This observation corroborates the assertion that self-refinement is instrumental in augmenting the proficiency in trivia creative writing tasks. Furthermore, the enhancement of single agents via self-refinement plays a pivotal role in fortifying the integrity of the overarching multi-agent framework.
> > > >
> > > > __Enhancing multi-agent collaboration through collaborative refinement action__.
> > > > For collaborative refinement, the process resembles the collaborative dialogue mentioned above, which involves integrating knowledge from different domains to accomplish tasks that demand cross-domain knowledge fusion. The results in the above table demonstrate the performance when collaborative refinement is absent. It's observable that compared to the scenario with AutoAgents, there is a decline of 2\%. Since the necessity for multiple agents to collaborate on a single task is entirely dependent on the decision made by the agent in the drafting phase, not all problems necessarily involve tasks that require collaborative refinement. However, it's evident that when this principle is omitted from the prompt's design, there's a noticeable performance decrease in AutoAgents. This also corroborates the beneficial impact of collaborative refinement in tackling complex tasks.
> > > >
> > > > -----
> > > >
> > > > > These three elements - the number of agents, self-refinement, and Collaborative Refinement Action - are key characteristics of this study from Table 1. Can you please explain why the OPEN-ENDED QUESTION ANSWER and The Trivia Creative Writing tasks are well-suited for assessing the impact of these factors?
> > > >
> > > > In the preceding section, we succinctly discussed the effects of self-refinement and collaborative refinement. In this part, our attention shifts to delving into the impact exerted by the number of agents, with a particular emphasis on underscoring the significance of dynamic agent generation in our system.
> > > >
> > > > __Dynamic agents enhance the adaptability of complex tasks__.
> > > > The ability to generate dynamic agents for various tasks is crucial for enhancing their adaptability to diverse scenarios. Figure 8 in Appendix A illustrates the contrast between GPT4 and AutoAgents’ responses to open-ended questions. Unlike GPT-4, AutoAgents can produce agents from three distinct domains, which can provide more elaborate answers. For the trivia creative writing task in Figures 9 and 10 in Appendix A, AutoAgents decomposed the task into four steps: first, it retrieved the answer to the question through a domain-specific agent and then constructed the story. Meanwhile, it is evident that the Language Expert Agent performed multiple checks on the coherence between the story content and the question, ensuring the accuracy of the story content. These examples all demonstrate the versatility of building a dynamic agent generation framework for complex tasks.
> > > >
> > > > -----

---

> > > > > ### Author Response · Authors · 2023-11-21
> > > > > **Thanks for your review!**
> > > > >
> > > > > __We appreciate your time in reading our feedback and look forward to further discussion and your suggestions.__

---

> ### Author Response · Authors · 2023-11-22
> **Official Comment by Authors**
>
> As the rebuttal phase draws to a close, we want to thank the reviewers for their valuable feedback. We believe we have successfully addressed your concerns in our responses and welcome any further questions you might have.
>
> Given the improvements and clarifications made based on your feedback, we kindly ask the reviewers to reconsider their evaluations to account for these updates. We greatly appreciate your consideration.

---

> > ### Comment · Reviewer_ypGp · 2023-12-03
> >
> > Thank you for your responses. I appreciate the ablation studies you conducted during the response period. As a result, I have decided to increase my score.

---

### Official Review · Reviewer_Tafn · 2023-10-31

**Soundness:** 2 fair
**Presentation:** 2 fair
**Contribution:** 2 fair
**Rating:** 5
**Confidence:** 3

**Summary:**

This paper propose AutoAgents, a framework to generate multiple agents and let them cooperate to solve different problems. The framework consists of the draft stage and the execution stage. The draft stage uses 3 predefined agents to cooperatively produce an agent list and an execution plan for a specific problem. The execution stage uses the proposed expert agents to execute the plan and solve the problem. Experiments on two benchmark show the effectiveness of the proposed framework.

**Strengths:**

- Clear presentation of high-level idea: the overall framework and process is clearly presented through well-drawn figures like Fig. 1 and 2.
- Strong reproducibility: the author provides source code and the temperature of LLM is set to 0, which makes it easy to reproduce the result in the paper.

**Weaknesses:**

- Limited novelty: according to Table 1, the main difference between the proposed framework and other existing methods like Social Simulacra, Epidemic Modeling, SSP, and AgentVerse is that this work uses self-refinement and collaborative refinement. This difference is more of a prompting technique and has already been used in many existing works like [1, 2, 3]
- Unclear presentation of detailed techniques: though the high-level idea is well-presented, the details of many technique are unclear. For example, how to determine when and which agents should engage in collaborative refinement? This is the main differnce from other methods but there is very little detailed description. More questions are in the Question part.
- Insufficient evaluation:
    1. Lack of baselines: Table 1 lists 12 existing frameworks, but none is used as baseline in task 1, and only 1 is used in task 2. Comparisons with existing methods are needed to show the effectiveness of the proposed methods.

    2. Lack of ablation study: there is no quantitive ablations on different components of the framework like self-refinement, collaborative refinement, etc.

    3. Unfair comparisons and potential problem in metric: in task 1, it is unfair to compare AutoAgents using GPT-4 with ChatGPT and Vicunna-13B. In task 2, the metric only considers the QA quality, how about the quality of the story around the given topic?

Reference:

[1] Noah Shinn, Beck Labash, and Ashwin Gopinath. Reflexion: an autonomous agent with dynamic memory and self-reflection. arXiv preprint arXiv:2303.11366, 2023.

[2] Zhibin Gou, Zhihong Shao, Yeyun Gong, Yelong Shen, Yujiu Yang, Nan Duan, and Weizhu Chen. Critic: Large language models can self-correct with tool-interactive critiquing, 2023.

[3] Wenlong Huang, Fei Xia, Ted Xiao, Harris Chan, Jacky Liang, Pete Florence, Andy Zeng, Jonathan Tompson, Igor Mordatch, Yevgen Chebotar, et al. Inner monologue: Embodied reasoning through planning with language models. arXiv preprint arXiv:2207.05608, 2022.

**Questions:**

Detailed techniques:

1. Section 3.1 Plan Generation: why plan genreation is parallel to agent generation? If the agent list has not been generated, how is it possible to "entail a clear identification of agent" for each step?

2. Section 3.2 Action Observer: how does the Action Observer interact and communicate with other agents to act as the tasks manager and how is this process determined? When will the Action Observer adapt the execution plan.

3. Section 3.2 Collaborative Refinement: how to determine when and which agents should engage in collaborative refinement?

4. Section 3.2 Knowledge Sharing Mechanism: how to determine what knowledge is shared and who to share with?

Experiments:

5. Compare with more baselines in Table 1 for both task 1 and task 2, especially multi-agent frameworks like Social Simulacra, Epidemic Modeling, SSP, and AgentVerse.

6. Unfair comparisons in Section 4.1: fair comparisons would be AutoAgents using ChatGPT v.s. ChatGPT and AutoAgents using Vicuna-13B v.s. Vicuna-13B.

7. Metric problem in Section 4.2: the trivia creative writing task has two subtask: (a) craft a coherent story around a given topic (2) answer N questions. The current metric only evaluate the result of subtask (b), there need another metric for subtask (a).

8. Ablation study: uantitive ablation results on different components of the framework. For example, remove self-refinement, collaborative refinement, different Observers, etc.

---

> ### Author Response · Authors · 2023-11-19
> **Official Comment by Authors**
>
> Thank you for your comments and suggestions. We think there might be a misunderstanding of our contributions. We will clarify these concerns with detailed responses in the following.
>
> ## Method
> > Limited novelty: according to Table 1, the main difference between the proposed framework and other existing methods like Social Simulacra, Epidemic Modeling, SSP, and AgentVerse is that this work uses self-refinement and collaborative refinement. This difference is more of a prompting technique and has already been used in many existing works like [1, 2, 3]
>
> Here, we further elucidate the main contributions of AutoAgents, a novel framework that addresses two salient issues: multi-agent generation and multi-agent cooperation. With respect to multi-agent generation, our method surpasses four existing methods in terms of the completeness and adaptability of agent creation, by employing Agent Generalization by Multi-Agent Discussion and Prompt Generalization. Regarding multi-agent cooperation, our framework exhibits the efficacy of Self-refinement Action and Cooperative Refinement Action. As illustrated in the table below, the distinctive features of AutoAgents encompass Agent Generalization by Multi-Agent Discussion, Prompt Generalization, Self-refinement Action, and Cooperative Refinement Action. AutoAgents is the first holistic framework that integrates multi-agent generation and collaboration. Detailed accounts of Agent Generalization by Multi-Agent Discussion and Prompt Generalization are given in below.
>
> | Method | Task | Agent Generalization by Multi-Agent Discussion | Prompt Generalization | Self-refinement Action | Collaborative Refinement Action |
> | --- | --- | --- | --- | --- | --- |
> | Social Simulacra | Social Simulation | &cross; | &cross; | &cross; | &cross; |
> | Epidemic Modeling | Social Simulation | &cross; | &cross; | &cross; | &cross; |
> | SSP | General Autonomous Agents | &cross; | &cross; | &cross; | &cross; |
> | AgentVerse | General Autonomous Agents | &cross; | &cross; | &cross; | &cross; |
> | AutoAgents | General Autonomous Agents | &check; | &check; | &check; | &check; |
>
> - __Agent Generalization by Multi-Agent Discussion__: Existing agent generation frameworks rely more on a single agent to generate the list of agents, but this often limits their adaptability to different tasks. For AutoAgents， the drafting stage is very important for the auto-generation of agents, as the list of agents to be generated is determined through cooperative discussions between the Planner agent and the two Observers. Additionally, in Appendix A, we further supplement with ablation experiments concerning the Agent Observer and Plan Observer, with the following table providing a detailed comparison. It is evident that in the absence of cooperative discussions among Observers, there is a significant decline in the overall performance of AutoAgents. This also validates the importance of generating reasonable agents for task adaptability.
>
> | Method | N (# triva questions) = 5|
> | --- | --- |
> | AutoAgents w/o observers | 87.0  |
> | AutoAgents | 90.0 |
>
> __Note__: Please refer to the Appendix A of the revision for the detailed setup of the experiment.
>
> - __Prompt Generalization__: The prompt employed by AutoAgents exhibits a more universal nature, signifying its capacity to acclimate to diverse tasks without necessitating bespoke customization. Both AgentVerse[1] and SSP[2] have implemented task-specific enhancements for varied task evaluations. In contrast, our methodology leverages a singular, unified prompt format to accommodate an array of tasks. The commendable efficacy in open-ended question answer and trivia creative writing tasks further corroborates the wide-ranging applicability and versatility of prompt design within the AutoAgents framework.
>
> __Note__: In the revision, we have improved Table 1 and provided additional comparisons in Appendix A.
>
> [1] https://github.com/OpenBMB/AgentVerse/tree/minecraft/agentverse/tasks
>
> [2] https://github.com/MikeWangWZHL/Solo-Performance-Prompting/tree/main/prompts
>
> -----

---

> > ### Author Response · Authors · 2023-11-19
> > **Official Comment by Authors**
> >
> > > 1. Unclear presentation of detailed techniques: though the high-level idea is well-presented, the details of many technique are unclear. For example, how to determine when and which agents should engage in collaborative refinement? This is the main differnce from other methods but there is very little detailed description.
> > > 2. Section 3.2 Collaborative Refinement: how to determine when and which agents should engage in collaborative refinement?
> >
> >
> > The implementation of collaborative refinement is entirely determined by discussions between the Planner and Observer. We have incorporated the principles of collaborative refinement into the Planner's prompt, without imposing restrictions on when to employ the collaborative refinement mode; this is left to the Planner's discretion. The specific content of the collaborative refinement prompt is as follows:
> > ```
> > 4.2 Each step should assign at least one expert role to carry it out. If a step involves multiple
> > expert roles, you need to specify the contributions of each expert role and how they collaborate
> > to produce integrated results.
> > ```
> >
> > -----
> >
> > > Section 3.1 Plan Generation: why plan genreation is parallel to agent generation? If the agent list has not been generated, how is it possible to "entail a clear identification of agent" for each step?
> >
> > In addressing your observation, it is important to clarify that the generation of agents and plans within our system is not conducted in parallel. The creation process of agents always precedes the development of the plan. To provide further clarity, we have added detailed explanations of the underlying principles for each prompt in the appendix, specifically outlining the process within the Planner.
> > ```
> > 1. Understanding and Breaking Down Tasks: The first step involves comprehensively understanding, analyzing, and deconstructing the given task or problem.
> >
> > 2. Utilization of Existing Expert Roles:
> > - Fully leverage existing expert roles suited to the problem.
> > - Ensure that these roles have cooperative or dependent relationships.
> > - Output the details of selected roles in JSON format, including their original information.
> >
> > 1. Creation of New Expert Roles:
> > - Avoid duplication of functions in new roles.
> > - New roles should have clear names, detailed descriptions, domain expertise, available tools, execution suggestions, and prompt templates.
> > - Ensure each new expert role has a distinct responsibility and domain of expertise.
> > - Specify the goals, constraints, and toolset for each new role.
> > - Provide execution suggestions and develop prompt templates for each new role.
> > - Output details of new roles in JSON format, following a specific structure.
> >
> > 1. Creation of Detailed Execution Plan:
> > - Develop a comprehensive plan with multiple steps addressing the problem.
> > - Assign at least one expert role to each step, detailing their contributions and collaborations.
> > - Provide detailed descriptions for each step, including expected outputs and inputs for subsequent steps.
> > - Include a final independent step for a language expert to provide a detailed response to the user's question.
> > - Present the execution plan as a numbered list, indicating the expert roles involved in each step.
> > ```
> >
> > To elaborate, the Planner's methodology is structured in a sequential manner: it begins with a thorough analysis of the problem at hand. Following this, it proceeds to generate a list of agents, tailored specifically to the identified problem. The final step involves the crafting of an execution plan. It is noteworthy, however, that agents and execution plans are inherently interdependent components. In recognition of this symbiosis, we have integrated both elements into the prompt's structure. This integrative approach facilitates the production of not only a more logical and targeted list of agents but also a more streamlined and effective execution plan.
> >
> > -----

---

> > > ### Author Response · Authors · 2023-11-19
> > > **Official Comment by Authors**
> > >
> > > > Section 3.2 Action Observer: how does the Action Observer interact and communicate with other agents to act as the tasks manager and how is this process determined? When will the Action Observer adapt the execution plan.
> > >
> > > In our manuscript, it is elucidated that the Action Observer adopts a vertical communication strategy for interfacing with agents. This entails assigning diverse tasks to different agents, guided by the specifics outlined in the execution plan, which includes both the identification of agents and their corresponding tasks. The Action Observer, in its role, meticulously reviews the results archived in long-term memory while also contemplating forthcoming execution strategies. This process is pivotal for assessing whether alterations to the plan are necessary. In instances where the extant outcomes are insufficient for the subsequent agent's requirements, the Action Observer takes the initiative to refine the execution plan, tailoring it based on these existing results. Presented below is a sample prompt that exemplifies the task modification process.
> > >
> > > ```
> > > 2.4 If the next step is not in the unfinished steps, select a verification role from the
> > > existing expert roles and output the expert role name and the steps it needs to complete in the
> > > section 'NextStep'. Please indicate the name of the expert role used at the beginning of the step.
> > > ```
> > > -----
> > >
> > > > Section 3.2 Knowledge Sharing Mechanism: how to determine what knowledge is shared and who to share with?
> > >
> > > To further elucidate, we delineate the specific agent types that are compatible with the three modes of knowledge sharing, as depicted in Figure 7 of our revised manuscript:
> > >
> > > - __Short-term Memory__: This is predominantly utilized by agents that emerge from self-refinement and collaborative-refinement processes. These agents leverage short-term memory to retain historical records throughout the refinement phase. Notably, every agent generated within our framework actively employs short-term memory.
> > >
> > > - __Long-term Memory__: This component archives the execution results of each agent, primarily chronicling the cumulative progress of the execution plan. Access to long-term memory is exclusive to a pre-designated Action Observer. The Action Observer utilizes this memory to evaluate whether additional execution steps are necessary or if there's a need to generate dynamic memory content.
> > >
> > > - __Dynamic Memory__: This memory type is essential for distilling and summarizing pivotal information from long-term memory for diverse agents. Generated by the Action Observer, dynamic memory serves to amalgamate critical aspects of the overall execution plan, thereby facilitating the agents responsible for implementing subsequent tasks. Consequently, the dynamic memory content available to each generated agent may differ, contingent upon their specific roles and tasks.
> > >
> > > -----

---

> > > > ### Author Response · Authors · 2023-11-19
> > > > **Official Comment by Authors**
> > > >
> > > > ## Experiemnts
> > > >
> > > > > 1. Lack of baselines: Table 1 lists 12 existing frameworks, but none is used as baseline in task 1, and only 1 is used in task 2. Comparisons with existing methods are needed to show the effectiveness of the proposed methods.
> > > > > 2. Compare with more baselines in Table 1 for both task 1 and task 2, especially multi-agent frameworks like Social Simulacra, Epidemic Modeling, SSP, and AgentVerse.
> > > > > 3. Unfair comparisons in Section 4.1: fair comparisons would be AutoAgents using ChatGPT v.s. ChatGPT and AutoAgents using Vicuna-13B v.s. Vicuna-13B.
> > > > > 4. Unfair comparisons and potential problem in metric: in task 1, it is unfair to compare AutoAgents using GPT-4 with ChatGPT and Vicunna-13B.
> > > >
> > > > Due to the specialized focus of Social Simulacra and Epidemic Modeling on Social Simulation tasks, a direct comparison with open-ended question-answering models is not practical. Furthermore, the operation of SSP is contingent upon an Azure GPT-4 key, which we currently do not possess. This limitation precludes us from acquiring results for task 1. As a result, we have enhanced Table 1 by including a comparative analysis featuring AgentVerse, to provide a more comprehensive overview within the scope of our available resources.
> > > >
> > > > In the revised Table 2, we have enhanced the analysis by including results for AgentVerse, utilizing both FairEval and HumanEval for a comprehensive evaluation. As illustrated in the table provided below, it is apparent that AutoAgents exhibits superior performance compared to AgentVerse in addressing a majority of open-ended questions. This improved efficacy can be attributed to several key features of AutoAgents: the implementation of cooperative discussion-based agent generation, and the integration of both self-refinement and collaborative refinement mechanisms. These components collectively contribute to the more effective handling of open-ended queries by AutoAgents, as demonstrated in the comparative results.
> > > >
> > > > | Method | AutoAgents vs. AgentVerse |
> > > > | --- | --- |
> > > > | FairEval | 91.3%  |
> > > > | HumanEval | 77.5% |
> > > >
> > > > __Note__: The prompt configuration within AgentVerse is tailored to its [brainstorming](https://github.com/OpenBMB/AgentVerse/blob/main/agentverse/tasks/tasksolving/brainstorming/config.yaml) task, incorporating modifications to upgrade the model to GPT-4.
> > > >
> > > > -----

---

> > > > > ### Author Response · Authors · 2023-11-19
> > > > > **Official Comment by Authors**
> > > > >
> > > > > > 1. In task 2, the metric only considers the QA quality, how about the quality of the story around the given topic?
> > > > > > 2. Metric problem in Section 4.2: the trivia creative writing task has two subtask: (a) craft a coherent story around a given topic (2) answer N questions. The current metric only evaluate the result of subtask (b), there need another metric for subtask (a).
> > > > >
> > > > >
> > > > > Here, we present a sample of task 2 where the topic is a noun with expansive semantic breadth. However, assessing the alignment between the narratives and these broad noun topics is less straightforward due to their general nature.
> > > > > ```
> > > > > {
> > > > >     "questions": ["Who was the man behind The Chipmunks?", "Which Lloyd Webber musical premiered in the US on 10th December 1993?", "Who was the next British Prime Minister after Arthur Balfour?", "Who had a 70s No 1 hit with Kiss You All Over?", "What claimed the life of singer Kathleen Ferrier?"],
> > > > >     "answers": [["David Seville", "david seville"], ["Sunset Blvd", "West Sunset Boulevard", "Sunset Boulevard", "Sunset Bulevard", "Sunset Blvd.", "sunset boulevard", "sunset bulevard", "west sunset boulevard", "sunset blvd"], ["Sir Henry Campbell-Bannerman", "Campbell-Bannerman", "Campbell Bannerman", "Sir Henry Campbell Bannerman", "Henry Campbell Bannerman", "Henry Campbell-Bannerman", "henry campbell bannerman", "sir henry campbell bannerman", "campbell bannerman"], ["Internal exile", "Exiles", "Transported for life", "Exile (politics and government)", "Voluntary exile", "Sent into exile", "Exile and Banishment", "Self-exile", "Forced exile", "Exile", "Exile in Greek tragedy", "Banish", "Banishment", "exiles", "voluntary exile", "forced exile", "banish", "self exile", "exile politics and government", "exile in greek tragedy", "sent into exile", "banishment", "transported for life", "exile", "internal exile", "exile and banishment"], ["Cancer pathology", "Deaths by cancer", "Anti-cancer", "Cancer (disease)", "Cancerophobia", "Malignant lesion", "Cancer medication", "Malignant tumors", "Cancer signs", "Malignant neoplasm", "Invasive (cancer)", "Malignant Neoplasms", "Malignant growth", "Sporadic cancer", "Malignant cancer", "Tumour virus", "Cancer en cuirasse", "Microtumor", "Malignant neoplasms", "Malignant tumour", "Carcinophobia", "Malignacy", "Cancer patient", "Epithelial cancers", "Solid cancer", "Cancers", "Tumor medication", "Malignant neoplastic disease", "AIDS-related cancer", "Invasive cancer", "Cancer therapy", "Cancerous tumor", "Cancer", "Financial toxicity", "Cancer diagnosis", "Cancer (medicine)", "Malignant tumor", "Cancerous", "Borderline (cancer)", "Signs of cancer", "Malignancies", "Cancer aromatase", "aids related cancer", "sporadic cancer", "cancer disease", "malignant tumors", "cancers", "carcinophobia", "cancer", "cancer diagnosis", "malignant neoplastic disease", "malignant neoplasm", "tumour virus", "cancer medicine", "deaths by cancer", "malignant tumour", "epithelial cancers", "solid cancer", "cancerous", "borderline cancer", "invasive cancer", "anti cancer", "cancer pathology", "cancer signs", "cancer aromatase", "cancer therapy", "financial toxicity", "cancerophobia", "cancer en cuirasse", "cancer patient", "cancerous tumor", "malignant cancer", "malignant neoplasms", "tumor medication", "signs of cancer", "malignacy", "malignant tumor", "cancer medication", "microtumor", "malignancies", "malignant lesion", "malignant growth"]],
> > > > >     "question_ids": ["tc_2", "tc_33", "tc_40", "tc_49", "tc_56"],
> > > > >     "topic": "Harry Potter"
> > > > > }
> > > > > ```
> > > > > To address task (a), we merged the data's answers with the topics to form new topics. We then utilize an LLM (BAAI/bge-large-en-v1.5) [1] to generate embeddings for these new topics and the stories, computing the similarity between them to gauge topic relevance. The results, as shown below, indicate that AutoAgents still exhibit superior relevance to the topic in this scenario.
> > > > >
> > > > > | Method | N (# triva questions) = 5| N (# triva questions) = 10|
> > > > > | --- | --- | --- |
> > > > > | SSP | 0.643 | 0.623 |
> > > > > | AutoAgents | 0.651 | 0.664 |
> > > > >
> > > > > [1] Zhang, Peitian, et al. "Retrieve Anything To Augment Large Language Models." arXiv preprint arXiv:2310.07554 (2023).
> > > > >
> > > > > -----

---

> > > > > > ### Author Response · Authors · 2023-11-19
> > > > > > **Official Comment by Authors**
> > > > > >
> > > > > > > 1. Ablation study: uantitive ablation results on different components of the framework. For example, remove self-refinement, collaborative refinement, different Observers, etc.
> > > > > > > 2. Lack of ablation study: there is no quantitive ablations on different components of the framework like self-refinement, collaborative refinement, etc.
> > > > > >
> > > > > > Within Appendix A of our revision, we have incorporated an ablation study specifically focused on AutoAgents within the context of trivia creative writing task. The table provided below delineates the performance fluctuations of AutoAgents when certain key components are omitted, thereby illustrating the impact of each individual element on the overall efficacy of the system.
> > > > > > - __AutoAgents w/o observers__: During the Drafting Stage, the Planner in AutoAgents engages in collaborative discussions with two Observers to determine the optimal list of agents and the execution plan. Table below elucidates that in the absence of observers, there is a marked 3% reduction in the overall performance of AutoAgents. This substantiates the imperative role of collaborative discussions in agent generation.
> > > > > > - __AutoAgents w/o self-refinement__: The performance of AutoAgents decreases by 3% in the absence of the self-refinement agent. This observation corroborates the assertion that self-refinement is instrumental in augmenting proficiency in trivia creative writing tasks. Furthermore, the enhancement of single agents via self-refinement plays a pivotal role in fortifying the integrity of the overarching multi-agent framework.
> > > > > > - __AutoAgents w/o collaborative refinement__: It's observable that compared to the scenario with AutoAgents, there is a decline of 2% without collaborative refinement. Since the necessity for multiple agents to collaborate on a single task is entirely dependent on the decision made by the agent in the drafting phase, not all problems necessarily involve tasks that require collaborative refinement. However, it's evident that when this principle is omitted from the prompt's design, there's a noticeable performance decrease in AutoAgents. This also corroborates the beneficial impact of collaborative refinement in tackling complex tasks.
> > > > > > - __AutoAgents w/o dynamic memory__: Dynamic memory predominantly addresses the requisites of specialized agents. The Action Observer amalgamates pivotal data for forthcoming tasks, utilizing the historical action records archived in long-term memory. The below table elucidates a 1% diminution in the efficacy of AutoAgents bereft of dynamic memory. Quintessential insights derived from dynamic memory are primarily assimilated into the prompt, thereby augmenting the comprehension of critical information and bolstering the operational proficiency of actions.
> > > > > >
> > > > > > | Method | N (# triva questions) = 5|
> > > > > > | --- | --- |
> > > > > > | AutoAgents w/o observers | 87.0 |
> > > > > > | AutoAgents w/o self-refinement | 87.0 |
> > > > > > | AutoAgents w/o collaborative refinement | 88.0  |
> > > > > > | AutoAgents w/o dynamic memory | 89.0  |
> > > > > > | AutoAgents | 90.0 |
> > > > > >
> > > > > > -----

---

> > > > > > > ### Author Response · Authors · 2023-11-21
> > > > > > > **Thanks for your review!**
> > > > > > >
> > > > > > > __We appreciate your time in reading our feedback and look forward to further discussion and your suggestions.__

---

> > > > > > > ### Comment · Reviewer_Tafn · 2023-11-21
> > > > > > >
> > > > > > > Thank you for answering my questions. The response has helped me better evaluate the contribution of the work. I'll increase my score to 5, but I still have some reservations about the work.
> > > > > > >
> > > > > > > 1. Unfair comparison. As said in my review, it is unfair to compare AutoAgents using GPT-4 with ChatGPT or Vicuna-13B. The performance improvement could come from the strong ability of GPT-4 instead of using AutoAgents. The fair comparison is AutoAgents using ChatGPT v.s. ChatGPT, and AutoAgents using Vicuna-13B v.s. Vicuna-13B. Moreover, since the human evaluators are asked to rate the score instead of comparing two outputs, I think it is better and more straightforward to report the average score instead of win rate.
> > > > > > > 2. Unclear detailed techniques. Although the authors have provided further explanations and prompts, these descriptions are still abstract and hard for the readers to understand or reproduce the details in these techniques based on the manuscript. Concrete examples to show the process of collaborative refinement or the functionality of the action observer could also improve the clarity, but there is no such example.

---

> > > > > > > > ### Author Response · Authors · 2023-11-21
> > > > > > > > **Thank you for your valuable and insightful feedback!**
> > > > > > > >
> > > > > > > > We greatly appreciate your insightful feedback. In response to your latest query, we have provided a detailed answer below.
> > > > > > > >
> > > > > > > > -----
> > > > > > > >
> > > > > > > > > Unfair comparison. As said in my review, it is unfair to compare AutoAgents using GPT-4 with ChatGPT or Vicuna-13B. The performance improvement could come from the strong ability of GPT-4 instead of using AutoAgents. The fair comparison is AutoAgents using ChatGPT v.s. ChatGPT, and AutoAgents using Vicuna-13B v.s. Vicuna-13B.
> > > > > > > >
> > > > > > > > Thank you for your valuable feedback. Following your recommendation, we have revised our manuscript to exclude the comparison between ChatGPT and Vicua-13B. This decision aligns with our commitment to ensuring a fair and balanced evaluation. We will now focus our comparative analysis solely on AutoAgents, GPT-4, and AgentVerse. For a detailed overview of these changes, please refer to the updated version of our manuscript or following table. We appreciate your guidance in enhancing the quality and integrity of our work.
> > > > > > > >
> > > > > > > > | Evaluator | vs. GPT-4 | vs. AgentVerse |
> > > > > > > > | --- | --- | --- |
> > > > > > > > | FairEval | 76.3% | 91.3%  |
> > > > > > > > | HumanEval | 62.5% | 77.5% |
> > > > > > > >
> > > > > > > > -----
> > > > > > > >
> > > > > > > > > Moreover, since the human evaluators are asked to rate the score instead of comparing two outputs, I think it is better and more straightforward to report the average score instead of win rate.
> > > > > > > >
> > > > > > > > We acknowledge that our initial description might have lacked clarity. To clarify, each volunteer in our study was presented with two distinct responses simultaneously for evaluation, as opposed to scoring a single response. This methodology ensured a more comprehensive and comparative analysis. An illustrative example of a volunteer’s scoring process is provided for reference.
> > > > > > > >
> > > > > > > > ```
> > > > > > > > Overall Evaluation Performance:
> > > > > > > > Assistant 1 performed way better than Assistant 5
> > > > > > > >
> > > > > > > > Assistant 1: 9.8 / 10
> > > > > > > > Assistant 5: 8.5 / 10
> > > > > > > >
> > > > > > > > Overall Evaluation Evidence:
> > > > > > > > - Assistant 1 provides a more structured and comprehensive response, covering a wider range of techniques, tools, and strategies in a well-organized format, making it highly helpful, relevant, and detailed but Assistant 1 tends to give irrelavant details to questions.
> > > > > > > >
> > > > > > > > - Assistant 5, while also relevant and accurate, offers a less broader, less structured approach with helpful insights, which may not be enough for the cerntain questions asked.
> > > > > > > >
> > > > > > > > Question 1: How can I improve my time management skills?
> > > > > > > >
> > > > > > > > Response for Assistant 1:
> > > > > > > > Helpful: 10
> > > > > > > > Relevant: 10
> > > > > > > > Accurate: 10
> > > > > > > > Level of Detail: 10
> > > > > > > > Average Score: 10
> > > > > > > >
> > > > > > > > Response for Assistant 5:
> > > > > > > > Helpful: 8
> > > > > > > > Relevant: 8
> > > > > > > > Accurate: 9
> > > > > > > > Level of Detail: 8
> > > > > > > > Average Score: 8.25
> > > > > > > > ```
> > > > > > > >
> > > > > > > > We regret any confusion caused by our previous statements and have accordingly revised the relevant sections in our manuscript to enhance clarity and understanding. We appreciate your attention to this matter and hope that these modifications address your concerns effectively.
> > > > > > > >
> > > > > > > > ```
> > > > > > > > For HumanEval, we enlisted three independent volunteers to evaluate two sets of responses—one generated by AutoAgents and the other by a different model—based on criteria such as helpfulness, reliability, accuracy, and comprehensiveness. Notably, the volunteers were blinded to the identity of the model that produced each response, ensuring an unbiased assessment.
> > > > > > > > ```
> > > > > > > >
> > > > > > > > -----
> > > > > > > >
> > > > > > > > > Unclear detailed techniques. Although the authors have provided further explanations and prompts, these descriptions are still abstract and hard for the readers to understand or reproduce the details in these techniques based on the manuscript. Concrete examples to show the process of collaborative refinement or the functionality of the action observer could also improve the clarity, but there is no such example.
> > > > > > > >
> > > > > > > > We appreciate your observations regarding the clarity and detail of our technical descriptions. We assure you that we have diligently incorporated all crucial technical details and logical constructs into the prompt, situated within the MetaGPT environment. This careful integration underpins the development of a comprehensive and automated system. As such, our manuscript encompasses all vital reproducible elements, offering substantial details for accurately replicating the algorithms we have introduced.
> > > > > > > >
> > > > > > > > However, recognizing the merit of your suggestions, we have taken proactive steps to further elucidate our methodologies. We have now included concrete examples that illustrate the process of collaborative refinement and the functionality of the action observer. These examples have been detailed in Figure 7 and Figure 9 within the revised appendix of our manuscript.
> > > > > > > >
> > > > > > > > -----
> > > > > > > >
> > > > > > > > We are confident that these additions significantly improve the clarity of our work and address the concerns you have raised regarding the reproducibility and understanding of our techniques.

---

> ### Author Response · Authors · 2023-11-22
> **Official Comment by Authors**
>
> As the rebuttal phase draws to a close, we want to thank the reviewers for their valuable feedback. We believe we have successfully addressed your concerns in our responses and welcome any further questions you might have.
>
> Given the improvements and clarifications made based on your feedback, we kindly ask the reviewers to reconsider their evaluations to account for these updates. We greatly appreciate your consideration.

---

### Official Review · Reviewer_uXv7 · 2023-11-01

**Soundness:** 2 fair
**Presentation:** 2 fair
**Contribution:** 3 good
**Rating:** 6
**Confidence:** 3

**Summary:**

This paper introduces a framework for automatically synthesizing collaborative specialized agents. AutoAgents mimics the collaborative process of human teams by decomposing tasks into drafting and execution phases and delegating different subtasks to different agents. AutoAgents couples the relationship between tasks and roles by dynamically generating multiple required agents based on task content and planning solutions for the current task based on the generated expert agents. Finally, the experimental and empirical evaluation on various benchmarks validates the advantages of AutoAgents.

**Strengths:**

1.	This paper presents a framework that adaptively generates and coordinates multiple specialized agents to build an AI team according to different tasks.
2.	The paper is technically sound and the research question is clear.
3.	The contribution of the paper is relevant for LLM-based multi-agent collaboration. The results of this paper is interesting and significant in automatic agent generation. The proposed AutoAgents framework generates more coherent and accurate solutions than the existing multi-agent methods.

**Weaknesses:**

1.	How the proposed AutoAgents framework expands the scope of collaborative applications and reduces the consumption of resources should be elaborated.
2.	The authors do not explain how to determine the number of agents in the section of the framework for automatic agent generation.
3.	The section about automatic agent generation is too tedious to introduce too much related works
4.	In addition to ChatGPT, Vicuna-13B and GPT4 in Table 2, it has not enough recent models to further show the superiority of the proposed framework-AutoAgents in open-ended question answer task in the experimental part.
5.	In the experimental part, the performance on N=10 is better than N=5 in trivia creative writing task, but there is no explanations.

**Questions:**

1.	During the execution stage, why the authors adopt the vertical communication paradigm, which assigns different responsibilities to agents according to their roles.
2.	The authors present results for the Open-ended Question Answer task and the Trivia Creative Writing task to evaluate the framework effectiveness. What if the Question Answer task is not open-ended? Does the proposed framework AutoAgents still work?

---

> ### Author Response · Authors · 2023-11-19
> **Official Comment by Authors**
>
> Thank you for your motivating review and concrete suggestions. Detailed responses to your questions are listed as follows.
>
> > __Q1:__ How the proposed AutoAgents framework expands the scope of collaborative applications and reduces the consumption of resources should be elaborated.
>
> Existing agent technologies are overly dependent on pre-defined prompts. This reliance poses a substantial user barrier and limits their practicality for broader applications. AutoAgents addresses this by automating the creation of new agent roles, which streamlines both the definition of roles and the execution of tasks, thereby enhancing the system's versatility.
>
> AutoAgents significantly extends its adaptability to diverse applications through the automatic generation of multiple intelligent agents, each tailored for specific tasks. This improvement in agent generation quality is achieved through the collaborative efforts of three predefined agents - the Planner, Agent Observer, and Plan Observer. These agents work together to discuss, refine, and finalize the list of agents and their corresponding execution plans. In Appendix A, we delve deeper into this aspect with ablation experiments focusing on the Agent Observer and Plan Observer. Our comparative analysis, as shown in the following table, reveals a marked decrease in AutoAgents' overall performance in the absence of cooperative dialogues among Observers. This outcome not only underscores the efficacy of collaborative refinement but also highlights the critical role of these discussions in enhancing task adaptability and resource optimization in the AutoAgents framework.
>
> | Method | N (# triva questions) = 5|
> | --- | --- |
> | AutoAgents w/o observers | 87.0  |
> | AutoAgents | 90.0 |
>
> __Note__: Please refer to the Appendix A of the revision for the detailed setup of the experiment.
>
> -----
>
> > __Q2:__ The authors do not explain how to determine the number of agents in the section of the framework for automatic agent generation.
>
> AutoAgents fundamentally predefines only a Planner and three types of Observers. Throughout the automatic agent generation phase, neither the number nor the types of agents to be generated are pre-specified. Instead, the decision regarding both the number and types of agents is dynamically and autonomously determined through collaborative discussions among the Planner, Agent Observer, and Plan Observer. This process ensures that the agents generated are optimally aligned with the specific requirements and nuances of each task.
>
> -----
>
> > __Q3:__ The section about automatic agent generation is too tedious to introduce too much related works
>
> In Appendix A, we have included an expanded comparison of methodologies for agent generation. Traditional frameworks in agent generation typically hinge on the capability of a single agent to determine the list of agents. This approach, however, often results in limited adaptability across various tasks. In contrast, the AutoAgents framework places significant emphasis on the drafting stage for the auto-generation of agents. Here, the compilation of the agent list is a result of collaborative discussions between the Planner agent and the two Observers, enhancing adaptability and flexibility.
>
> Moreover, the prompts utilized in AutoAgents are designed with a universal approach, reflecting their ability to adapt to a wide range of tasks without the need for task-specific customization. While platforms like AgentVerse[1] and SSP[2] have opted for task-specific modifications, AutoAgents adopts a more holistic, unified prompt strategy. This uniform approach is adept at catering to diverse tasks, which is evident in our system's notable performance in open-ended question answering and trivia creative writing tasks. Such results highlight the extensive applicability and adaptability of our prompt design, making AutoAgents a versatile tool in the realm of agent-based systems.
>
> | Method | Task | Agent Generalization by Multi-Agent Discussion | Prompt Generalization |
> | --- | --- | --- | --- |
> | Social Simulacra | Social Simulation | &cross; | &cross; |
> | Epidemic Modeling | Social Simulation	| &cross; | &cross; |
> | SSP | General Autonomous Agents | &cross; | &cross; |
> | AgentVerse | General Autonomous Agents | &cross; | &cross; |
> | AutoAgents | General Autonomous Agents | &check; | &check; |
>
> [1] https://github.com/OpenBMB/AgentVerse/tree/minecraft/agentverse/tasks
>
> [2] https://github.com/MikeWangWZHL/Solo-Performance-Prompting/tree/main/prompts
>
> -----

---

> > ### Author Response · Authors · 2023-11-19
> > **Official Comment by Authors**
> >
> > > __Q4:__ In addition to ChatGPT, Vicuna-13B and GPT4 in Table 2, it has not enough recent models to further show the superiority of the proposed framework-AutoAgents in open-ended question answer task in the experimental part.
> >
> > In the revised Table 2, we have enhanced the analysis by including results for AgentVerse, utilizing both FairEval and HumanEval for a comprehensive evaluation. As illustrated in the table provided below, it is apparent that AutoAgents exhibits superior performance compared to AgentVerse in addressing a majority of open-ended questions. This improved efficacy can be attributed to several key features of AutoAgents: the implementation of cooperative discussion-based agent generation, and the integration of both self-refinement and collaborative refinement mechanisms. These components collectively contribute to the more effective handling of open-ended queries by AutoAgents, as demonstrated in the comparative results.
> >
> > | Method | AutoAgents vs. AgentVerse |
> > | --- | --- |
> > | FairEval | 91.3%  |
> > | HumanEval | 77.5% |
> >
> > __Note__: The prompt configuration within AgentVerse is tailored to its [brainstorming](https://github.com/OpenBMB/AgentVerse/blob/main/agentverse/tasks/tasksolving/brainstorming/config.yaml) task, incorporating modifications to upgrade the model to GPT-4.
> >
> > -----
> >
> > > __Q5:__ In the experimental part, the performance on N=10 is better than N=5 in trivia creative writing task, but there is no explanations.
> >
> > Table 3 clearly reveals that all evaluated methods perform more effectively at N=10 than at N=5. This enhanced performance at N=10 can be ascribed to the wider array of available resources, which considerably assists the methods in crafting narratives that are both more cohesive and realistic. In contrast, the constrained resource pool at N=5 means that the exclusion of even a single response category can result in significant fluctuations in performance. This is exemplified by a marked 20% performance drop due to the lack of one type of response. At N=10, however, such performance disparities are notably mitigated, evidenced by a more modest 10% change under similar conditions of response type absence. This observation underscores an interesting aspect for more detailed exploration in our future research.
> >
> > -----
> >
> > > __Q6:__ During the execution stage, why the authors adopt the vertical communication paradigm, which assigns different responsibilities to agents according to their roles.
> >
> > Our approach is grounded in the belief that specific tasks are best handled by specialized Agents to achieve the most effective results. This principle advocates for assigning a single Agent to address a distinct set of challenges within its expertise, rather than diluting its focus across multiple domains. In situations where a task requires a solitary response, an agent within our vertical structure is specifically appointed to fulfill that task. This targeted allocation of responsibilities ensures that each agent operates within its area of specialization, thereby optimizing performance. This strategy, which is also employed in the AgentVerse framework, underscores the benefits of harnessing the strengths of specialized agents for specific domains, validating our approach in the vertical communication model.
> >
> > -----
> >
> > > __Q7:__ The authors present results for the Open-ended Question Answer task and the Trivia Creative Writing task to evaluate the framework effectiveness. What if the Question Answer task is not open-ended? Does the proposed framework AutoAgents still work?
> >
> > The Trivia Creative Writing task serves as a rigorous test for large language models, gauging their ability to retrieve and integrate a wide range of information from their internal knowledge bases. In this task, models are required to construct a cohesive narrative around a specified topic while seamlessly incorporating answers to N trivia questions. The evaluation criteria for this task consider any match to the answer variants of a question as a correct mention.
> >
> > Consequently, the Trivia Creative Writing task fits the description of a non-open-ended question-answering challenge. In this setup, each sample presents a series of either 5 or 10 common sense questions that necessitate accurate responses from diverse methodologies before the models proceed to narrative creation. The principal measure for assessing this task is the successful integration of these correct answers into the narratives. An analysis of the results in Table 3 clearly demonstrates that narratives generated by AutoAgents feature a notably higher proportion of accurate responses. This outcome robustly validates the capability of AutoAgents in the sphere of non-open-ended question-answering, underlining their adeptness at effectively amalgamating precise information within imaginative storylines.

---

> > ### Comment · Reviewer_uXv7 · 2023-11-23
> >
> > Thank the authors for their feedback. After reading other reviewers' comments and the authors' rebuttal, I decide to maintain the score as is.

---

> > > ### Author Response · Authors · 2023-11-23
> > > **Thank you!**
> > >
> > > Thank you for your practical advice and quick feedback, which have significantly improved our paper and enhanced its impact. We greatly appreciate your positive evaluation.

---

> ### Author Response · Authors · 2023-11-21
> **Thanks for your review!**
>
> __We appreciate your time in reading our feedback and look forward to further discussion and your suggestions.__

---

> ### Author Response · Authors · 2023-11-22
> **Official Comment by Authors**
>
> As the rebuttal phase draws to a close, we want to thank the reviewers for their valuable feedback. We believe we have successfully addressed your concerns in our responses and welcome any further questions you might have.
>
> Given the improvements and clarifications made based on your feedback, we kindly ask the reviewers to reconsider their evaluations to account for these updates. We greatly appreciate your consideration.

---

### Official Review · Reviewer_paZk · 2023-11-02

**Soundness:** 2 fair
**Presentation:** 2 fair
**Contribution:** 3 good
**Rating:** 6
**Confidence:** 4

**Summary:**

The paper proposes AutoAgents, a framework to generate and coordinate multiple specialised agents with distinct roles to construct an AI team to accomplish specialised tasks. The process comprises two stages: Drafting and Execution. The drafting stage involves Planner, Agent Observer, and Plan Observer agents discussing to generate the agent team and an execution plan, which is executed by the generated agents in the execution stage. The authors evaluated the performance of AutoAgents against a few existing solutions in the Open-ended Question-answering and Trivia Creative Writing task, and results show that AutoAgents performs better against the tested baselines. They performed a qualitative evaluation in a task requiring AutoAgents to generate the Tetris game.

**Strengths:**

The idea of dynamically generating agents who play different roles to solve team tasks is interesting and useful. I found the idea to be novel. It is easy for the reader to get a good overview of the idea of AutoAgents. However, there was a need to look at supplementary materials to understand aspects of what the different predefined roles were supposed to do. The visuals helped me understand the idea better. The background was sufficient, in my opinion, and well-written. This discussion and Table 1 made the contributions clear.

**Weaknesses:**

Section 3:
For the agent generation, the motivation for the format of the Prompt P is unclear. Additionally, when we look at the supplementary material, the specific elements of the prompt are not explained -- are these taken from existing works?

Others:
I also found details that needed to be included in a few other sections, such as the self-refinement process. Furthermore, I had questions about specific choices of parameters during the evaluations. I have included my questions in the next part to capture the specific places where I needed more information.

Minor typos:
Page 2: effectiveness of AutoAgents. [we] also conduct
Page 7: at = lt ∪ pt ∪ ot, [where lt,] where lt denotes

**Questions:**

1) What motivated the design of the prompt elements for the predefined agents? Did you consider alternatives, or did existing works inspire these?
2) How were the roles, skills, and actions decided for the specific tasks? Were they injected in the prompt, or did the Planner agent generate them?
3) For the self-refinement process, what was the source of the thoughts, i.e., who decided what thoughts to include and why?
4) Regarding knowledge sharing, did you experiment with each agent using different types, or were these predefined onset?
5) How did you decide to use the number of discussions in the two stages to 3 and 5, respectively (page 7)?
6) In multiple places, the agents' discussions may be stopped after some predefined threshold if the agents do not reach a consensus (e.g. during collaborative refinement, page 7). How often did this happen, and if the team did process, what is the quality of the outcome?
7) In the Open-ended Q&A, the authors mention recruiting volunteers, but no more details are provided about how, whether ethics approval was sought, etc. Could the authors please provide more details on this.

**Details Of Ethics Concerns:**

In the Open-ended Q&A, the authors mention recruiting volunteers, but no more details are provided about how, whether ethics approval was sought, etc.

---

> ### Author Response · Authors · 2023-11-19
> **Official Comment by Authors**
>
> We thank you for your constructive comments! We have fixed the grammatical errors in the revision. We also have addressed the other comments below and incorporated the feedback in the revision.
>
> -----
>
> > __Q1:__ What motivated the design of the prompt elements for the predefined agents? Did you consider alternatives, or did existing works inspire these?
>
>
> Existing agent technologies are overly dependent on pre-defined prompts, presenting a significant barrier to users and rendering them impractical for widespread use. By developing a system that automatically creates new agent roles, we can automate both the definition of roles and the execution of tasks, significantly increasing the technology's versatility. AutoAgents, potentially a pioneering framework, is distinctively geared towards generating universal prompts applicable to a wide range of agents. Our design not only ensures the accomplishment of each agent's predefined tasks but also, through careful consideration during the design phase, enhances the adaptability of these prompts to a variety of tasks, thus broadening their general applicability. In the updated appendix, we have incorporated a section at the outset of each prompt detailing the design principles and underlying logic.
>
> -----
>
> > __Q2:__ How were the roles, skills, and actions decided for the specific tasks? Were they injected in the prompt, or did the Planner agent generate them?
>
> The Planner agent serves as the architect in our framework, initially crafting a comprehensive array of roles specifically designed for a variety of tasks. This includes not only the creation of prompts, descriptions, and toolsets for each role but also the formulation of an intricate execution plan. This plan distinctly specifies whether self-refinement or collaborative refinement actions are required. Following this initial phase, both the list of roles and the execution plan are subject to further refinement, achieved through collaborative dialogues with Observer agents. It is imperative to note that the choice of specific tools and techniques employed by each agent to navigate the steps in the execution plan is made independently by the agents themselves. A crucial aspect of our methodology is that we do not create unique prompts for each task. Rather, the primary prompts are methodically generated by the Planner agent, ensuring a consistent and streamlined approach across various tasks.
>
> -----
>
> > __Q3:__ For the self-refinement process, what was the source of the thoughts, i.e., who decided what thoughts to include and why?
>
> As described in the appendix about the 'custom agent', each agent, upon receiving their assigned task, must first understand and analyze the task. They need to plan the steps required for future action and determine the immediate steps to be taken. The agent then outputs the results of completing the current step. In the next round of feedback, the history of executed steps and their outcomes (short-term memory) are fed back into the prompt under 'completed steps and responses'. This helps the agent to determine the steps needed in the future. The prompt's design rationale is additionally detailed in the appendix of the revision.
>
> -----

---

> ### Author Response · Authors · 2023-11-19
> **Official Comment by Authors**
>
> > __Q4:__ Regarding knowledge sharing, did you experiment with each agent using different types, or were these predefined onset?
>
> To further elucidate, we delineate the specific agent types that are compatible with the three modes of knowledge sharing, as depicted in Figure 7 of the revision:
>
> - __Short-term Memory__: This is predominantly utilized by agents that emerge from self-refinement and collaborative-refinement processes. These agents leverage short-term memory to retain historical records throughout the refinement phase. Notably, every agent generated within our framework actively employs short-term memory.
>
> - __Long-term Memory__: This component archives the execution results of each agent, primarily chronicling the cumulative progress of the execution plan. Access to long-term memory is exclusive to a pre-designated Action Observer. The Action Observer utilizes this memory to evaluate whether additional execution steps are necessary or if there's a need to generate dynamic memory content.
>
> - __Dynamic Memory__: This memory type is essential for distilling and summarizing pivotal information from long-term memory for diverse agents. Generated by the Action Observer, dynamic memory serves to amalgamate critical aspects of the overall execution plan, thereby facilitating the agents responsible for implementing subsequent tasks. Consequently, the dynamic memory content available to each generated agent may differ, contingent upon their specific roles and tasks.
>
> Due to the fundamental nature of Short-term Memory and Long-term Memory as indispensable components within the system, our investigation was exclusively focused on understanding and analyzing the role of dynamic memory. In the Appendix A of the revision, we have included an ablation study of dynamic memory. The following table shows the results without the dynamic memory. It is observed that the performance of AutoAgents decreases by 1% without the summarization from dynamic memory, which also proves the importance of dynamic memory.
>
> | Method | N (# triva questions) = 5|
> | --- | --- |
> | AutoAgents w/o dynamic memory | 89.0  |
> | AutoAgents | 90.0 |
>
> __Note__: Please refer to the Appendix A of the revision for the detailed setup of the experiment.
>
> -----
>
> > __Q5:__ How did you decide to use the number of discussions in the two stages to 3 and 5, respectively (page 7)?
>
> On one hand, it's been observed through experience that a detailed plan and a complete list of agents can generally be obtained after about three rounds of collaborative planning discussions. In cases where a single agent is operating, there's a high probability that the current task can be completed within five rounds. While a greater number of execution rounds typically leads to better outcomes, on the other hand, since we rely on GPT-4, more iterations mean increased costs. This setup is designed to manage and control the costs associated with task execution.
>
> -----
>
> > __Q6:__ In multiple places, the agents' discussions may be stopped after some predefined threshold if the agents do not reach a consensus (e.g. during collaborative refinement, page 7). How often did this happen, and if the team did process, what is the quality of the outcome?
>
> The frequency of such occurrences is inherently tied to the complexity of the problem or task at hand. It is acknowledged that a premature termination of the refinement process could potentially compromise the quality of the final output. To address this, once the specified limit is reached, our system prompts with an instruction: 'You should synthesize the responses of previous steps and provide the final feedback.' This directive is designed to guide the agent towards concluding the refinement process, ensuring that a coherent and comprehensive outcome is still achieved despite the early halt in discussions.
>
> -----

---

> > ### Author Response · Authors · 2023-11-21
> > **Thanks for your review!**
> >
> > __We appreciate your time in reading our feedback and look forward to further discussion and your suggestions.__

---

> > ### Author Response · Authors · 2023-11-21
> > **Official Comment by Authors**
> >
> > > __Q7.__ In the Open-ended Q&A, the authors mention recruiting volunteers, but no more details are provided about how, whether ethics approval was sought, etc. Could the authors please provide more details on this.
> >
> > For HumanEval, we enlisted three independent volunteers to evaluate two sets of responses—one generated by AutoAgents and the other by a different model—based on criteria such as helpfulness, reliability, accuracy, and comprehensiveness. Notably, the volunteers were blinded to the identity of the model that produced each response, ensuring an unbiased assessment. We instructed the volunteers, who are responsible for assessing the quality of different feedback, to adhere to these standards.
> >
> > ```
> > We would like to request your feedback on the response to the user question
> > displayed above. Please rate the helpfulness, relevance, accuracy, level of
> > details of their responses.
> >
> > Each response receives an overall score on a scale of 1 to 10, where a higher
> > score indicates better overall performance. Please first provide a
> > comprehensive explanation of your evaluation, avoiding any potential bias and
> > ensuring that the order in which the responses were presented does not affect
> > your judgment.
> >
> > Output with the following format:
> > Evaluation evidence: <your evluation explanation here>
> > Score: <score>
> > ```

---

> ### Author Response · Authors · 2023-11-22
> **Official Comment by Authors**
>
> As the rebuttal phase draws to a close, we want to thank the reviewers for their valuable feedback. We believe we have successfully addressed your concerns in our responses and welcome any further questions you might have.
>
> Given the improvements and clarifications made based on your feedback, we kindly ask the reviewers to reconsider their evaluations to account for these updates. We greatly appreciate your consideration.

---

> > ### Comment · Reviewer_paZk · 2023-11-23
> >
> > Thank you for your clarifications and updates. I have updated my score.

---

> > > ### Author Response · Authors · 2023-11-23
> > > **Thank you!**
> > >
> > > Thank you for your practical advice and quick feedback, which have significantly improved our paper and enhanced its impact. We greatly appreciate your positive evaluation.

---

### Author Response · Authors · 2023-11-20
**Dear Reviewers (rebuttal)**

We express our gratitude to the reviewers for their insightful feedback!

__In this post__:
- We highlight key positive aspects noted in the reviews.
- We outline the revisions made in the updated PDF document.

__In the individual replies__, we address additional comments and concerns.

## Positive things
- __Innovation__
  - ```paZk```: "The idea of dynamically generating agents who play different roles to solve team tasks is interesting and useful. I found the idea to be novel."
  - ```uXv7```: "The contribution of the paper is relevant for LLM-based multi-agent collaboration. The results of this paper is interesting and significant in automatic agent generation. The proposed AutoAgents framework generates more coherent and accurate solutions than the existing multi-agent methods."
  - ```ypGp```: "In comparison to AgentVerse and SSP, this research stands out by highlighting the significance of Self-Refinement agents and Collaborative Refinement Action as key differentiators."
- __Clarity and Presentation__
  - ```paZk```: "The visuals helped me understand the idea better. The background was sufficient, in my opinion, and well-written. This discussion and Table 1 made the contributions clear."
  - ```uXv7```: "The paper is technically sound and the research question is clear."
  - ```Tafn```: "Clear presentation of high-level idea: the overall framework and process is clearly presented through well-drawn figures like Fig. 1 and 2."
  - ```ypGp```: "This study provides a valuable clarification of its position, especially in the context of LLM-based Agent frameworks."
- __Reproducibility__
  - ```Tafn```: "Strong reproducibility: the author provides source code and the temperature of LLM is set to 0, which makes it easy to reproduce the result in the paper."

## Changes to the PDF

### Main text

- ```[paZk]``` (Page 2) Minor capitalization correction..
- ```[uXv7]``` (Section 2.2) Enhanced wording contrasting AutoAgents with previous solutions.
- ```[Tafn]``` (Table 1) A more comprehensive comparison of agent generation methods.
- ```[paZk]``` (Page 7) An expanded description of the knowledge sharing mechanism.
- ```[uXv7,Tafn]``` (Section 4.1) A detailed performance comparison with AgentVerse.
- ```[paZk]``` (Section 4.1) A detailed explanation of HumanEval.


### Appendix
The content added based on the reviews:

- ```[Tafn,ypGp]``` (Section A) An ablation study covering observers, self-refinement, collaborative refinement, and dynamic memory.
- ```[uXv7,Tafn]``` (Table 5) A comparative analysis of existing frameworks and AutoAgents in multi-agent generation methods.
- ```[paZk]``` (Figure 8) A legend illustrating three knowledge-sharing mechanisms.
- ```[paZk]``` (Section B) A thorough explanation of HumanEval.
- ```[Tafn,ypGp]``` (Section D) Detailed design principles for predefined prompts.
- ```[Tafn]``` (Figure 7 & Figure9) Examples of collaborative refinement and the action observer.

We hope that these responses and revisions adequately address the points raised and enhance the clarity and impact of our work.

---

### Meta-Review · Area_Chair_EnB3 · 2023-12-06

**Metareview:**

This paper proposes a prompting technique for LLMs to generate responses by first prompting the LLM to generate multiple expert personalities during a "drafting" phase, followed by coordinating their responses during an "execution" phase. While the results of this approach are promising, the paper is missing important experimental details.

Most notably, the paper's results are based on prompting GPT-4, which is a black box and not reproducible. Secondly, the final Tetris experiment is missing crucial details around the experiment setup, e.g. how is success measured on this task and what is the success rate under this metric?

Moreover, since the paper requires chaining together the outputs of multiple prompts—each corresponding to its own prompt—the choice of baselines should be reconsidered. As AutoAgents uses multiple instances of prompting the LLM, methods like CoT that only prompt the LLM once are inherently disadvantaged in a way that is independent of the technique presented in this work. For example, a fairer comparison may be to prompt via CoT an equivalent number of times as AutoAgents, and taking the best output. Relatedly, the iteration limit of AutoAgents is not clearly specified for the experiments.

**Justification For Why Not Higher Score:**

The recommendation is based on the reasons stated in the meta-review: Mainly, this work in its current state cannot be easily reproduced, and secondly, the baseline comparisons seem poorly designed.

**Justification For Why Not Lower Score:**

N/A

---

### Decision · Program_Chairs · 2024-01-16

Reject